# Cu single-atom embedded g-C₃N₄ nanosheets rehabilitate multidrug-resistant bacteria infected diabetic wounds via photoswitchable cascade reaction

Xichen Sun[1,2,7], Pengqi Zhu[3,4,7], Liuyan Tang[1,2], Pengfei Wang[1,2], Ningning Li[1,2], Qing Wang[1,2], Yan-Ru Lou [5], Yuezhou Zhang [1,2] & Peng Li [1,2,6]

To tackle elevated blood glucose, multidrug-resistant (MDR) bacterial infections, and persistent inflammation in diabetic wounds, we present a therapeutic strategy that employs a photoswitch-controlled catalytic cascade reaction, utilizing a photocatalytic material engineered through the synergistic regulation of nitrogen vacancies and single-atom embedding. Under visible light illumination, the N vacancy exist in g-C₃N₄ (CN) significantly enhances photocatalytic glucose oxidation to regulate the hyperglycemia condition at diabetic wound sites, and the atomically dispersed Cu promotes the generation of •OH and •O₂⁻ to efficiently eliminate MDR bacteria ( > 99.9%). Under dark conditions, excess ROS are scavenged by Cu/CN, reducing inflammation of wounds and promoting polarization of M2 macrophages. Serum biochemical and vital organs histopathological analyses after 14 days of treatment confirm the biosafety profile of Cu/CN. This photoswitchable cascade reaction effectively treats MDR bacterial-infected diabetic wounds in male mice, highlighting its potential for antibiotic-free therapy with promising clinical translation applications.

Diabetes, along with its complications-particularly non-healing chronic wounds prone to bacterial infections-has emerged as a critical threat to public health, imposing a heavy burden on patients' life quality and social economy[1,2]. The diabetic wounds are characterized by excessive presence of glucose, persistent bacterial colonization, chronic inflammation, increased levels of oxidative stress, and abnormal angiogenesis, etc[3,4]. The rise of bacterial resistance has diminished the efficacy of conventional antibiotics, making bacterial infected diabetic wounds even harder to heal[5,6]. *Staphylococcus aureus* (Gram-positive) and *Escherichia coli* (Gram-negative) are the most frequent bacteria isolated from diabetic wounds, the fast emergence and spread of their multidrug-resistant (MDR) strains, such as methicillin-resistant *S. aureus* (MRSA) and extended-spectrum *β*-lactamases producing *E. coli* (ESBL *E. coli*), have intensified the demand for alternative antibacterial strategies[7]. For example, a case study revealed that MRSA accounted for more than half of *S. aureus* isolated from diabetic foot ulcers[8]. Strategies capable of effectively combating MDR bacteria, reducing glucose levels, controlling oxidative stress, regulating inflammation,

[1]State Key Laboratory of Flexible Electronics (LoFE) & Institute of Flexible Electronics (IFE), Frontiers Science Center for Flexible Electronics (FSCFE), Northwestern Polytechnical University, Xi'an, China. [2]Key Laboratory of Flexible Electronics of Zhejiang Province, Ningbo Institute of Northwestern Polytechnical University, Ningbo, China. [3]Shanxi Bethune Hospital, Shanxi Academy of Medical Sciences, Tongji Shanxi Hospital, Third Hospital of Shanxi Medical University, Taiyuan, China. [4]Tongji Hospital, Tongji Medical College, Huazhong University of Science and Technology, Wuhan, China. [5]Faculty of Pharmacy, University of Helsinki, Helsinki, Finland. [6]Research and Development Institute of Northwestern Polytechnical University, Shenzhen, China. [7]These authors contributed equally: Xichen Sun, Pengqi Zhu. ✉e-mail: iamyzzhang@nwpu.edu.cn; iampli@nwpu.edu.cn

and promoting angiogenesis are highly desired for the treatment of diabetic wounds.

Glucose oxidase (GOx) has been employed to treat diabetic complications by catalyzing the oxidative depletion of glucose—an essential nutrient for bacterial growth—and generating $H_2O_2$, which adversely affects bacterial viability[9–11]. Recently, a number of glucose oxidase-based platforms that effectively promoting diabetic wound healing have been developed[12–16]. In addition, nanozymes with the similar activity and wider applicability have been developed to overcome some limitations of natural enzymes. For example, metal-based nanozymes ($MnO_2$, Au, $TiO_2$, $Cu_2O$, and $CeO_2$, etc.) have been constructed to mimic GOx activity[17–19]. However, a challenge with these nanozymes is the lack of precise control over their reaction process, which typically results in non-stop catalysis and the production of excessive ROS at the wound site, thus exacerbating the inflammation and disturbing the wound healing process[20,21]. A potential solution strategy involves the use of switchable processing methods that can regulate enzyme activity under specific stimuli. Among various stimulation methods, light stands out due to its spatiotemporal selectivity, enabling precise regulation of the activity of photocatalysis, thus playing a key role in the controllable treatments[22]. For example, He and coworkers used visible light as a switch to activate photocatalytic titanium dioxide, which consumed glucose at the wound site to produce hydrogen and promote wound healing[23].

Graphitic carbon nitrides ($g$-$C_3N_4$) have demonstrated effective and controllable photocatalytic performance, making them widely used in energy, environmental, and biomedical fields[24–29]. $g$-$C_3N_4$ exhibits the ability to photocatalytically oxidize glucose to generate $H_2O_2$, and has been employed as a GOx-mimicking photocatalyst for glucose detection[30,31]. Defect engineering techniques[32–35] such as carbon and nitrogen (N) vacancies[36–39], as well as oxygen doping[40,41] in $g$-$C_3N_4$, can effectively improve its photocatalytic performance by altering the electronic band structure[42], optimizing carrier transfer[43], and enhancing surface active sites[44]. $g$-$C_3N_4$ typically exhibits a low adsorption rate for $H_2O_2$ and is unable to further consume the $H_2O_2$ produced in the previous step through cascade reaction. By loading metal single-atoms such as Ru[45], Pt[46], and Cu[47] onto $g$-$C_3N_4$, the resulting single-atom embedded $g$-$C_3N_4$ demonstrates enhanced photocatalytic cascade reaction performance, further consuming $H_2O_2$ to produce ROS that are more effective at killing bacteria. Recent studies have employed several single-atom catalysts for photocatalytic antibacterial applications, including Ag[48], Zn[49], and Cu[50] single atoms. However, none of these works explored the synergistic combination with N vacancies to enhance their photocatalytic activity. Furthermore, there is limited exploration into combining defect engineering with single-atom embedding to optimize the photocatalytic performance of $g$-$C_3N_4$ for the enhanced treatment of diabetic infected wounds.

Herein, we present a Cu single-atom embedded N vacancy-rich $g$-$C_3N_4$ photocatalyst (Cu/CN), which utilizes a cascade reaction controlled via photoswitchable treatment methods to effectively regulate glucose levels and antibacterial activity while preventing excessive ROS, thereby minimizes the risk of overtreatment and reduces inflammation at the wound site, ultimately enhancing chronic wound healing (z 1). The N vacancies enhance photocatalytic glucose oxidation, effectively reducing blood glucose at the wound site. HPLC analysis identifies arabinose as the primary reaction product, confirming that the photocatalytic mechanism of Cu/CN fundamentally differs from conventional glucose oxidase activity. Meanwhile, the •OH produced from the photocatalytic decomposition of $H_2O_2$ in the preceding step, combined with the •OH and •$O_2^-$ produced through Cu single-atom-mediated photocatalytic water splitting, effectively eliminate MDR bacteria. The synergistic effect of N vacancy and Cu single-atom was also verified through DFT. In addition, Cu/CN scavenge excess ROS in dark reaction to alleviate inflammation at the wound site, thereby promoting the transition of macrophages from pro-inflammatory M1 phenotype to anti-inflammatory M2 phenotype. This photoswitchable cascade reaction of Cu/CN thus represents a promising and innovative approach for treatment of chronic diabetic wounds.

## Results

### Morphology, size, and composition characterization of CN

As shown in Fig. 1, CN with graphite-phase structure was prepared by thermal polymerization using melamine as the precursor, and the above materials were improved. To introduce N vacancy, the material was calcined at different temperatures, decomposing some CN under an air atmosphere[51]. The SEM images (Fig. S1) and TEM images (Fig. S2) revealed that the surface of CN becomes increasingly wrinkled and porous as the calcination temperature rise. $N_2$ adsorption experiments further confirmed that higher calcination temperatures increase in surface area and pore size, potentially enhancing the material's capability to adsorb the reaction substrate (Fig. S3 and Table S1). XRD results (Fig. 2a) indicated that with increasing heat treatment temperature, the peak intensity of the material decreased at 13° (100) crystal plane (stacking of in-plane structural units) and 27° (002) crystal plane (stacking of interlayer structures), indicating a reduction in the crystallinity of carbon nitride. The XPS was employed to examine the chemical states of the samples, using the binding energy of the C 1$s$ line at 284.8 eV from alkyl or adventitious carbon as a reference. A comparison of the C 1$s$ and N 1$s$ spectra of $CN_{550}$, $CN_{600}$, $CN_{650}$, $CN_{700}$ and $CN_{750}$ is shown in Fig. 2b, c. Both $CN_{550}$ and $CN_{700}$ exhibited characteristic binding configurations of carbon and nitrogen that are typical of graphitic carbon nitride. Specifically, in addition to the peak observed at 284.8 eV, the C1$s$ spectrum for CN featured two additional peaks at 286.1 eV and 288.05 eV, which correspond to C-$NH_2$ and $C(N)_3$ moieties within the graphitic carbon nitride structure, respectively[52].

After heat treatment, the ultraviolet-visible absorption strength of CN was enhanced (Fig. 2d). The strong electron paramagnetic resonance (EPR) signal peak with a g value of 2.004 was clearly observed (Fig. 2e). As the heat treatment temperature rose, the N vacancies in CN first increased, peaking at $CN_{700}$, and then decreased. To further prove the N vacancies in different samples, we performed elemental analysis and Raman spectroscopy. The increased C/N mass ratio from 0.65 to 0.69 supported the formation of N vacancies (Fig. S4), and the decreased Raman peak intensities indicate the structural unit of the CN is compromised with the increase of annealing temperature (Fig. S5). FTIR provided further insight into these defects' states. The spectra obtained from the samples (Fig. 2f) displayed distinct peaks characteristic of graphitic carbon nitride. A peak around 810 $cm^{-1}$ was observed, indicating out-of-plane bending vibrations in heptazine ring systems. Additionally, peaks observed in the range of 900–1800 $cm^{-1}$ correspond to C−N(−C)−C vibrations within the heterocyclic structures and/or bridging C−NH−C units. Broad peaks spanning from 3000 $cm^{-1}$ to 3500 $cm^{-1}$ were attributed to the $NH/NH_2$ group. However, when the CN samples were subjected to rapid thermal treatment, marked changes were evident in the FTIR signals. A prominent peak appeared at 2175 $cm^{-1}$, corresponding to $-C \equiv N$ (cyano groups), and its intensity increased with higher treatment temperature. The emergence of these cyano groups disrupted the hydrogen bonds between the polymeric melon strands, causing layer fluctuation and the collapse of the periodic stacking structure, as supported by the XRD findings.

### VIS-photocatalytic performances

CN with varying concentrations of N vacancies were used to investigate the photocatalytic activity of glucose consumption, and it was found that $CN_{700}$, with the highest concentration of N vacancies, exhibited the best photocatalytic activity (Fig. 3a). Therefore, $CN_{700}$ was selected for loading Cu in subsequent experiments. Additionally, to assess the photocatalyst's capability, measurements of the transient

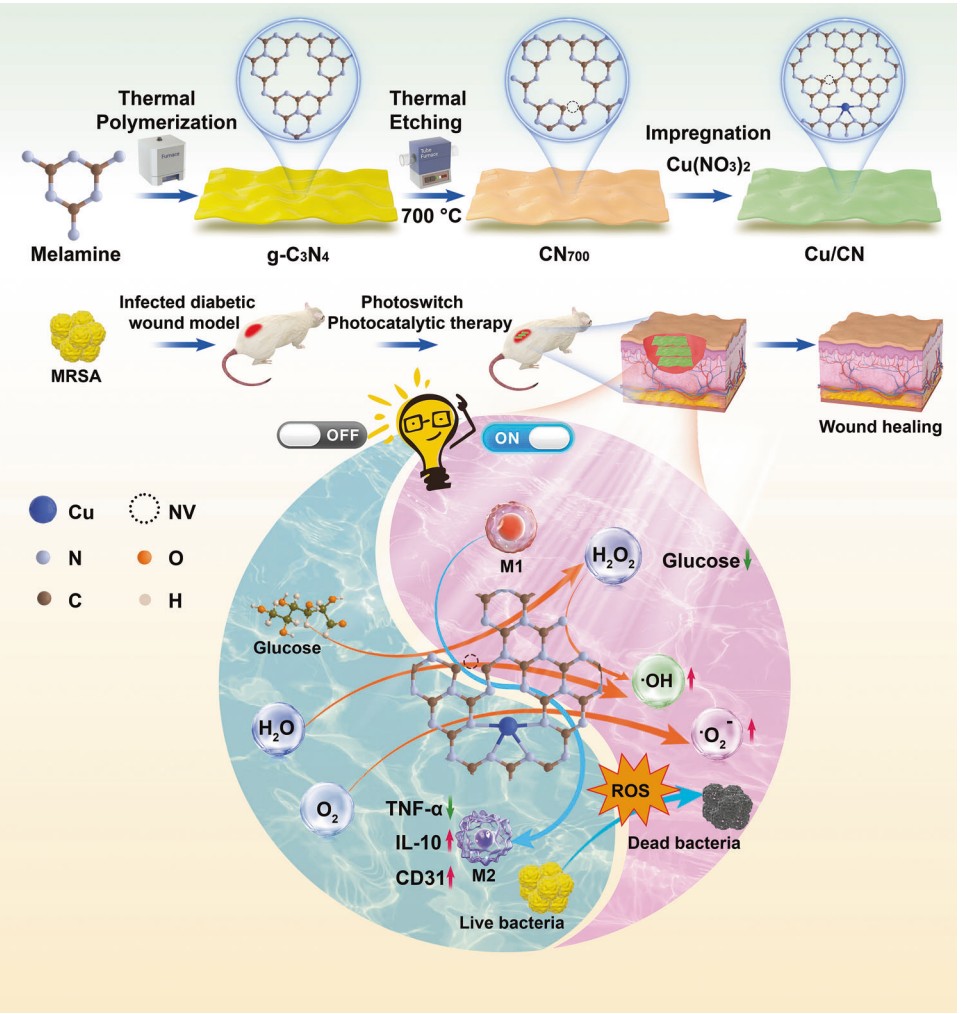

**Fig. 1 | Schematic diagram of the preparation of Cu/CN and the diabetic wound healing mechanism of Cu/CN.**

photocurrent responses were conducted. As shown in Fig. 3b, all samples exhibited reproducible and stable photocurrent signals, with photocurrent density directly proportional to the N vacancy density. $CN_{700}$ showed the highest photocurrent intensity, indicating a lower electron-hole pairs and improved migration efficiency. This was likely due to N vacancies acting as capture centers for photogenerated holes, which inhibited electron-hole recombination and promote separation, thus extending charge carrier lifetime[53,54]. Therefore, more photogenerated electrons and holes in $CN_{700}$ contributed to the photocatalytic consumption of glucose. Electron-transfer resistance was further investigated using electrochemical impedance spectroscopy (EIS). Typically, a smaller arc radius on the EIS Nyquist plot signifies reduced interfacial contact resistance between the electrode and the electrolyte[52]. As shown in Fig. 3c, the radius was inversely proportional to the N vacancy density, with $CN_{700}$ exhibiting the smallest radius of circularity, suggesting that its lower electron-transfer resistance facilitates carrier separation. According to $(Ah\nu)^2$ and the tangent intercept of the curve of light energy, the bandgap energy ($E_g$) was calculated. The $E_g$ values of $CN_{550}$, $CN_{600}$, $CN_{650}$, $CN_{700}$, and $CN_{750}$ were 2.8, 2.8, 2.77, 2.73, and 2.78 eV, respectively, indicating that the introduction of N vacancy slightly reduced the band gap of $CN_{700}$ and moderately increased its absorbance, consistent with UV-vis spectrum results. To further verify that the N vacancy modified samples enhance carrier separation, the charge carrier density ($N_d$) was evaluated using Mott-Schottky (M-S) curves. Figure 3e revealed that the M-S curve has a positive slope, which was characteristic of n-type semiconductors like

$CN_{700}$, allowing $N_d$ values to be calculated by Eq. (1). The $N_d$ values for the $CN_{550}$, $CN_{600}$, $CN_{650}$, $CN_{700}$, and $CN_{750}$ were estimated to be $1.8522 \times 10^{21}$, $2.3153 \times 10^{21}$, $2.3453 \times 10^{21}$, $3.2008 \times 10^{21}$, and $2.0185 \times 10^{21}$ cm$^{-3}$, respectively (Table S2). These $N_d$ values correlated with N vacancy density, with $CN_{700}$ having the highest $N_d$, indicating that N vacancies in the catalyst can increase the carrier density.

Moreover, the catavlyst's potential relative to the standard hydrogen electrode (SHE), was calculated from the Nernst Eq. (2) and the M-S curve, allowing determination of the conduction band (CB) position. The intersection of the tangent with the M-S curve's x-axis represented the catalyst's potential relative to the Ag/AgCl electrode; for n-type semiconductors, the CB position is 0.2 V lower than this potential. The CB positions of catalysts $CN_{550}$, $CN_{600}$, $CN_{650}$, $CN_{700}$, and $CN_{750}$ (relative to the SHE: −0.17, −0.19, −0.26, −0.19 and −0.37 V, respectively) were shown in Table S3. Combining these $E_g$ values with the corresponding catalysts' Tauc curves (Fig. 3f), the valence band potentials were determined to be 2.63, 2.61, 2.50, 2.54, and 2.40 V, respectively. Thus, as N vacancies increased, the bandgap width narrowed, consistent with the PBE calculation results (Fig. S6). This narrowing allowed $CN_{700}$ to utilize more low-energy photons, facilitating the photocatalytic consumption of glucose, which aligns with it observed the photocatalytic activity. To analyze the charge-carrier lifetime for different samples, we performed time-resolved photoluminescence (TRPL) spectroscopy. As presented in Fig. S7 and Table S5, the fluorescence lifetime of CN increases from 6.54 to 12.45 ns as the annealing temperature rises. This trend correlates with

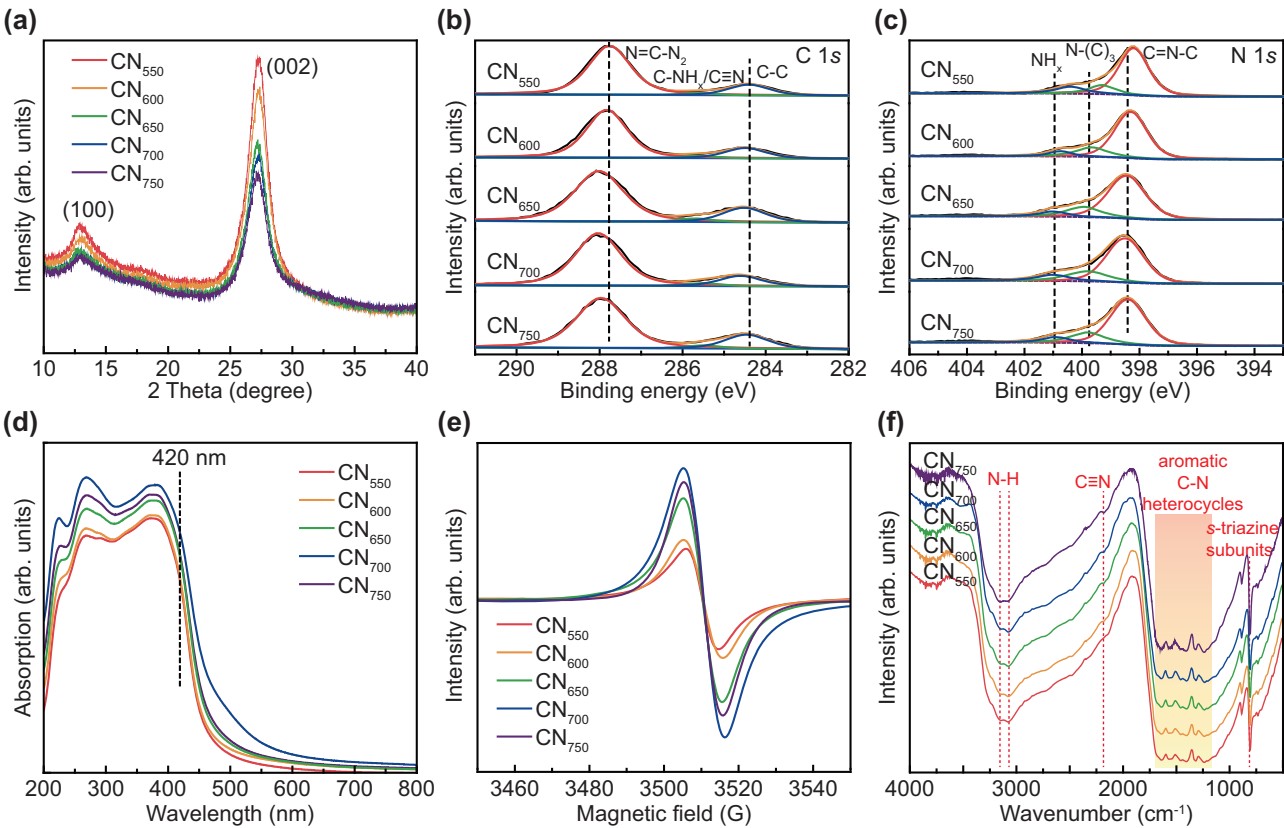

**Fig. 2 | Composition characterization of CN. a** XRD patterns, **b** C 1*s* XPS spectra, **c** N 1*s* XPS spectra, **d** UV–vis spectra, **e** EPR spectra and **f** FT-IR spectra of CN with different N vacancies concentrations.

the increase of N vacancies, which promotes the photogenerated carriers utilization capability of CN.

**Morphology, size, and composition characterization of Cu/CN**
CN rich in N vacancies was prepared by thermal etching, and atomically dispersed Cu on Cu/CN was synthesized using impregnation method. From the kinetics of $H_2O_2$ consumption by Cu/CN photocatalysis, it was found that 7.4 µg/L of $H_2O_2$ was consumed in 10 min (Fig. S8). The visible light absorption intensity of Cu/CN in the UV-vis spectrum was stronger than $CN_{700}$ (Fig. 4a), indicating that Cu/CN was more efficient in utilizing visible light for photocatalytic reaction, consistent with the PBE calculation results (Fig. S6). In addition, from the investigation of Cu/CN photocatalytic consumption of glucose under different wavelengths and light intensities, the photocatalytic activity of the Cu/CN exhibited positive correlations with both its UV-vis absorption capacity and the light intensity (Figs. S9, 10). The FT-IR spectra of Cu/CN showed a reduced intensity in C-N vibration within the heterocycle, possibly due to the influence of Cu single-atom on the vibration modes (Fig. 4b). The XRD patterns of Cu/CN showed no Cu crystal peaks (Fig. 4c), indicating that Cu single-atom does not disrupt the material's crystal structure. The Cu 2*p* XPS spectra of Cu/CN confirmed the presence of both +1 and +2 oxidation states (Fig. 4d)[55]. To further investigate the oxidation state of the Cu single-atom in Cu/CN, we analyzed X-ray absorption energy near-edge structure (XANES) of Cu foil, $Cu_2O$, CuO, and Cu/CN (Fig. 4e). The findings suggested that the XANES spectrum of Cu/CN falls between the spectra of $Cu_2O$ and CuO, indicating that the oxidation state of the Cu single-atom was between +1 and +2. From the slope and valence plot, the valency of Cu was estimated to be +1.75 (Fig. 4f). Extended X-ray absorption fine structure (EXAFS) was investigated to examine the coordination environment of the Cu sites. The first coordination shell, observed at 1.5 Å in Fig. 4g, suggested the presence of Cu–N or/and Cu–O

coordination. Quantitative EXAFS curve fitting analysis in R spaces was employed to characterize the coordination structure of the Cu atoms (Fig. 4h and Fig. S11). No significant Cu–Cu scattering was detected at 2.2 Å in Cu/CN samples, further confirming the atomically dispersed Cu. Wavelet transform of the EXAFS data distinguished backscattering atoms (Fig. 4i), revealing maximum intensities at 7.2 and 10.6 Å$^{-1}$ for Cu foil and CuO, respectively, which correspond to the Cu–Cu configuration. In contrast, Cu/CN displayed a maximum intensity at 4.5 Å$^{-1}$, attributed to Cu–N coordination. The N species produced from melamine during pyrolysis help fill vacancies, stabilizing the atomic Cu. EXAFS fitting results show the coordination number of Cu–N was approximately 4 (Table S4). By fitting the EXAFS spectra to different structural models that were optimized using DFT (Fig. 5), we found that the optimal fit was achieved when the Cu was coordinated with 4 N atoms, as shown in the atomic Cu structure model for Cu/CN in the inset of Fig. 1.

Furthermore, the observation by aberration-corrected high-angle annular dark-field scanning transmission electron microscopy (AC-HAADF-STEM) also verified the atomically dispersed Cu on Cu/CN nanosheets (Fig. 4j). And the energy-dispersive X-ray spectroscopy (EDS) elemental mapping confirmed the homogeneous distribution of constituent elements (C, N) and embedded copper (Cu) across the Cu/CN nanosheets (Fig. 4k). The actual content of Cu was 0.99 ± 0.01 wt% as determined by inductively coupled plasma optical emission spectroscopy (ICP-OES). We also measured the EPR spectra to analyze the in situ generation of •OH and •$O_2^-$ by Cu/CN with and without light exposure (Fig. S12). Under light, Cu/CN exhibited superior performance in photocatalytic decomposition of $H_2O_2$ to generate •OH compared to $CN_{700}$. Additionally, Cu/CN could directly photocatalyze water splitting to produce •OH and oxidize $O_2$ to produce •$O_2^-$, making it suitable for ROS generation for antibacterial purposes. To elucidate the photoswitch action, we performed in situ EPR analysis of Cu/CN

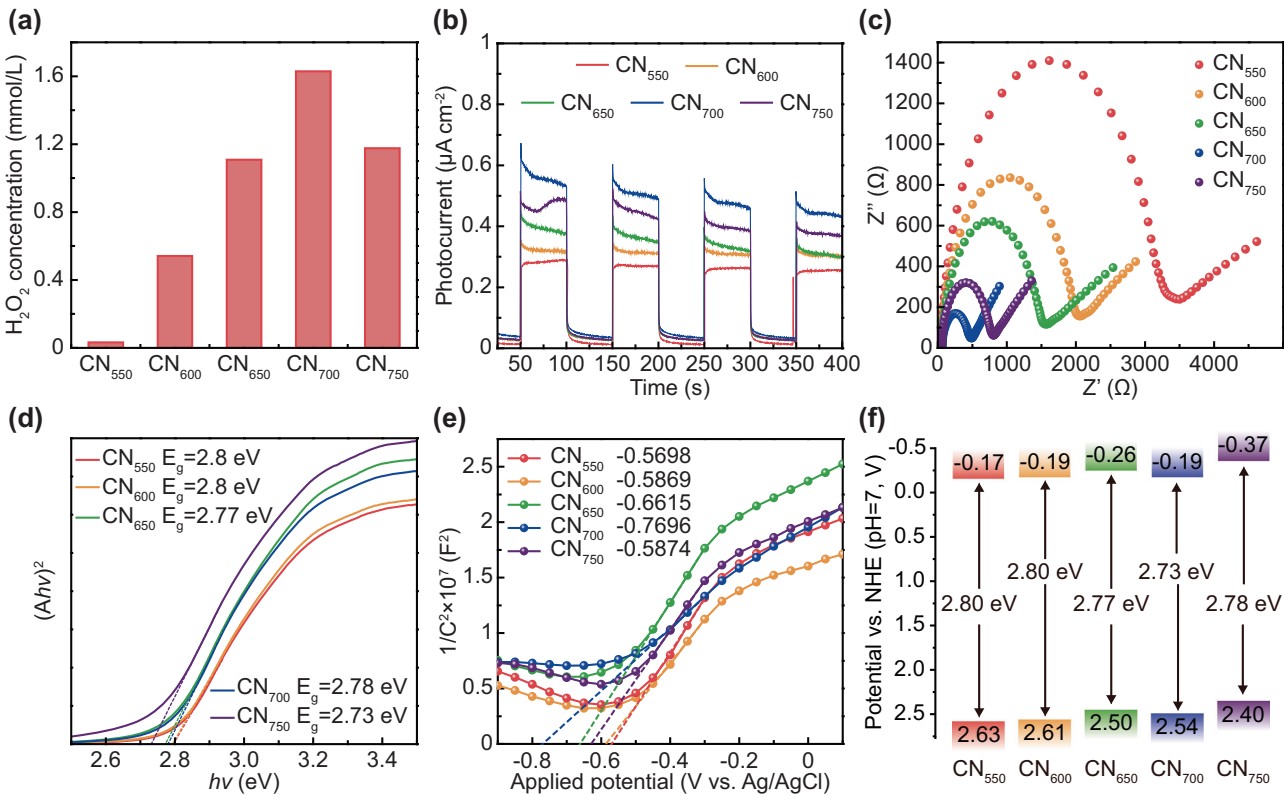

**Fig. 3 | Photoelectrochemical behaviors of modified CN. a** Photocatalytic properties of different catalysts for glucose consumption. **b** Transient photocurrent responses. **c** Electrochemical impedance spectra (EIS). **d** The plots of $(Ahv)^2$ and $hv$ of different catalysts. **e** M–S curves of different catalysts. **f** Schematic illustration of band structure for the different catalysts.

under cyclic illumination, alternating between light exposure and dark intervals. The in situ EPR results and semi-quantitative statistical analysis (Fig. S13, S14) demonstrated the light-dependent generation of •OH and •$O_2^-$, with no ROS observed under dark conditions, furthermore, detected no ROS generation under those low-pH conditions without light illumination (Fig. S15). Such results indicated that the generation of ROS in this system can be controlled by photoswitch. However, excessive ROS at the wound site could trigger pro-inflammatory cascades, ultimately hindering the healing process. To further explicate the ROS scavenging behavior of Cu/CN, we conducted DPPH and ABTS assays across varying pH conditions (3.0-7.0). As shown in Figs. S16, S17, Cu/CN exhibited marginally enhanced ROS scavenging activity at lower pH. Furthermore, intracellular ROS scavenging investigation using RAW 264.7 murine macrophages confirmed that Cu/CN effectively eliminates excess ROS in dark conditions, indicating its light-independent antioxidant activity (Fig. S18)[56,57]. XPS analysis of Cu 2$p$ in Cu/CN after ROS scavenging experiment revealed that the $Cu^{2+}$ content increased from 29.1% to 47.6% (Fig. S19), indicates there is a Cu(+1)/Cu(+2) redox cycle in the photocatalysis and scavenging processes[58–60].

## DFT of Cu/CN
Density functional theory (DFT) calculations were conducted to investigate the effects of N vacancies in CN and Cu single-atom in the adsorption energies of glucose and $H_2O_2$ on Cu/CN. For glucose, there were two types of adsorption sites on the CN surface: i) glucose adsorbs at N site with an adsorption energy of −0.15 eV, and ii) glucose adsorbs at N vacancy site with an adsorption energy of −0.42 eV (Fig. 5a). For $H_2O_2$, there were three types of adsorption sites on the Cu/CN surface of: i) $H_2O_2$ adsorbed at N vacancies, ii) $H_2O_2$ adsorbed between N vacancy and Cu single-atom sites, and iii) $H_2O_2$ adsorbed at

Cu single-atom, with corresponding adsorption energies of −0.26 eV, −0.74 eV and −1.39 eV, respectively (Fig. 5b). The more negative the adsorption energy, the more stable the adsorption, indicated that the presence of N vacancy enhance the catalyst's glucose adsorption capacity. Furthermore, charge transfer between CN and glucose was visualized using the Charge Density Difference (CDD) analysis. Figures S20, S21 demonstrate electron accumulation (yellow) at N site and electron depletion (blue) at N vacancy, revealing that N vacancy exhibit superior efficacy in facilitating glucose oxidation. The products of photocatalytic glucose consumption were analyzed by HPLC. The determined main product was arabinose, and there was no glucuronic acid found (Fig. S22). The complete equation was shown in illustration of (Fig. 5c (1)). The oxidation reaction occurring at the valence band position was shown in illustration of Fig. 5c (2)). The reduction reaction of $H_2O_2$ generated by single electron transfer of $O_2$ at the conduction band position was shown in illustration of (Fig. 5c (3) (4)). DFT calculations were performed for both reactions, with the corresponding energy profiles presented in Fig. 5c, d and Figs. S23, S24. The upstream step was endothermic and non-spontaneous, whereas the downstream step is exothermic and spontaneous. The step with the highest endothermic energy corresponded to the rate-determining step (RDS) of the overall reaction. A lower energy barrier for the RDS leads to enhanced catalytic activity. The transition state (TS) corresponds to a critical configuration where the glucose molecule undergoes C-C bond elongation prior to methanol elimination, representing a metastable "bond-breaking yet unbroken" state. DFT calculations reveal that the C-C bond length increases from 1.54 Å (ground state) to 1.60 Å in the presence of Cu/CN, comparatively, $CN_{700}$ induces greater bond elongation to 1.62 Å under identical conditions (Fig. S23). These results demonstrate that Cu/CN exhibits superior catalytic activity for hydroxymethyl elimination from glucose, as evidenced by the lower

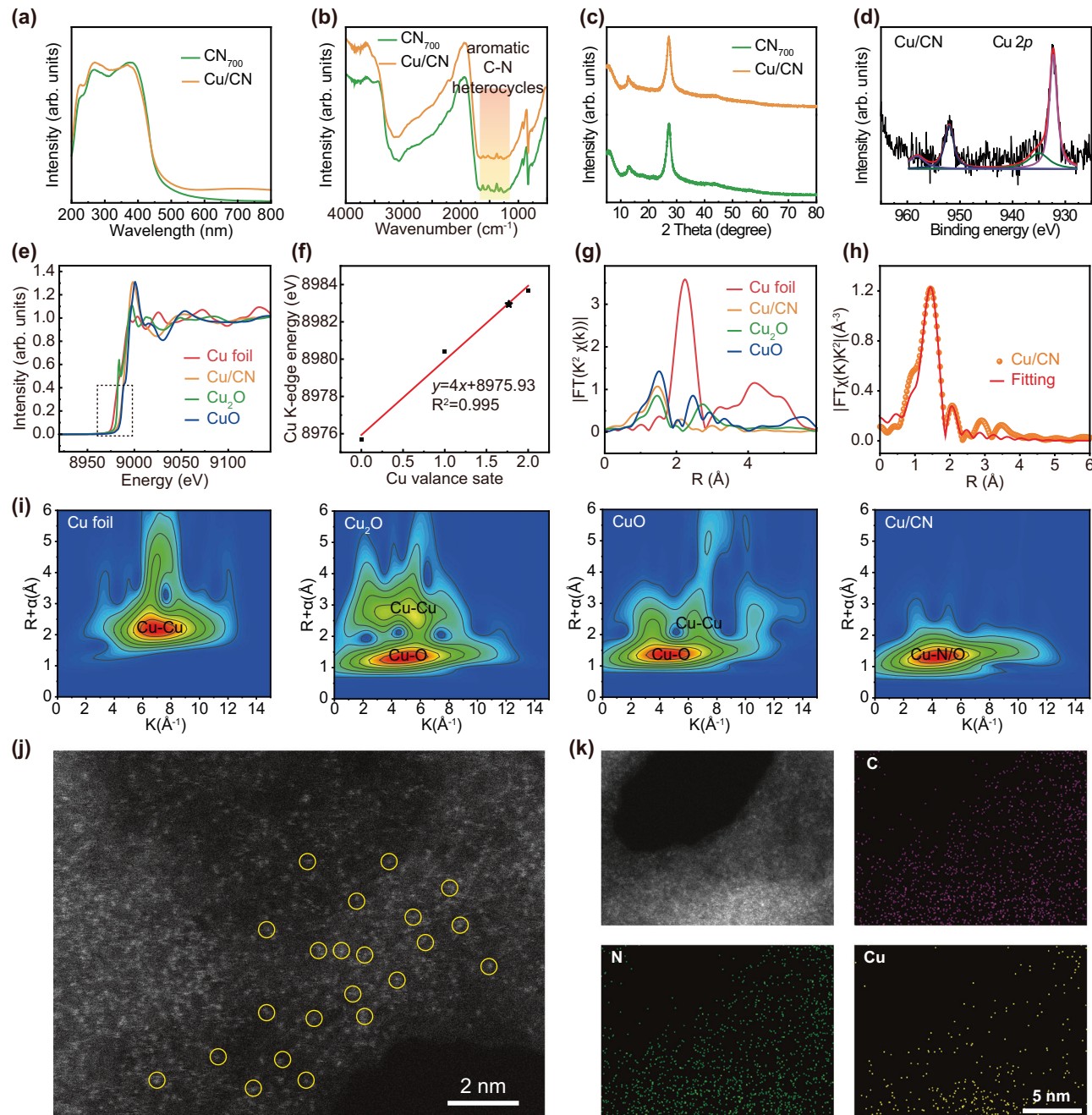

**Fig. 4 | Structural characterizations of Cu/CN. a** UV−vis spectra, **b** FT-IR spectra, **c** XRD patterns of $CN_{700}$ and Cu/CN. **d** Cu 2*p* XPS spectra of Cu/CN. **e** XANES spectra at Cu K-edge of Cu/CN. **f** Standard curve of Cu valence state. **g** EXAFS spectra at Cu K-edge of Cu/CN. **h** EXAFS fitting result of Cu/CN at R space. **i** Wavelet transform of Cu foil, $Cu_2O$, CuO and Cu/CN. **j** AC-HAADF-STEM image of Cu/CN (the atomically dispersed Cu was highlighted by yellow circles). **k** EDS mapping of Cu/CN. The experiments for (**j**, **k**) were repeated three times independently with similar results.

reaction energy barrier required to achieve the transition state compared to $CN_{700}$. The presence of Cu single-atom catalysts significantly reduced the energy barrier of the rate-determining step in reaction (2), thereby facilitating the reaction kinetics. For reaction (4), the Cu single-atom catalyst enabled the formation of $H_2O_2$ with the lowest energy barrier, enhancing the thermodynamic favorability of this pathway. Furthermore, the subsequent generation of •OH proceed with a relatively low activation energy barrier over the Cu single-atom sites, indicating a kinetically favorable process. Therefore, based on the experimental results (Fig. 3a) and charge density difference analysis (Figs. S20, S21), the presence of N vacancies facilitated the oxidation of glucose, while the Cu single-atom sites promote the formation of •OH radicals with enhanced reaction kinetics. DFT

calculations revealed that the embedding of Cu single-atom significantly reduces the reaction energy barrier in photocatalytic water splitting, thereby enhancing the reaction kinetics. The rate-limiting step involved the cleavage of the O−H bond, which serves as the key intermediate transition state in the process (Figs. 5e and S25).

### In vitro antibacterial property of Cu/CN

Motivated by the theoretical and experimental studies above, we hypothesized that Cu/CN could act as an effective bactericidal agent when glucose present via a photocatalytic cascade reaction. To validate this concept, we conducted an in vitro antibacterial study using ESBL *E. coli* and MRSA as model bacteria. The results, presented in Fig. 6a, c, demonstrate that the bactericidal rates for both strains were

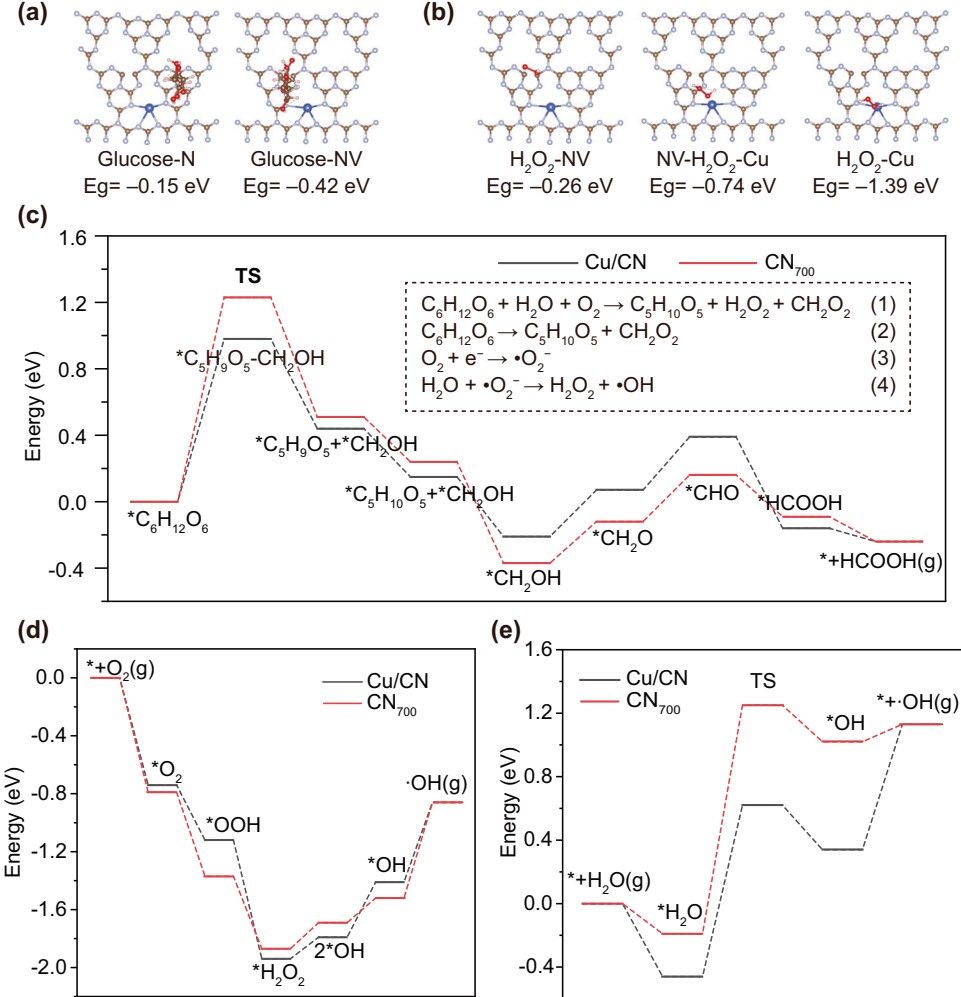

**Fig. 5 | DFT of Cu/CN. a** Adsorption energy of glucose at non vacancy and N vacancy sites on CN surface. **b** Adsorption energy of $H_2O_2$ at N vacancy, non-vacancy, and Cu single-atom sites on Cu/CN surface. The energy distribution corresponding to different transition states in (**c**) glucose conversion, **d** generation and conversion of $H_2O_2$, and **e** water splitting. The illustration in (**c**) shows the (1) complete equation, (2) oxidation reaction occurring at the valence band position, and (3) (4) reduction reaction of $H_2O_2$ generated by single electron transfer of $O_2$ at the conduction band position, respectively.

significantly higher when treated with Cu/CN+vis compared to the control, GOx, and $CN_{700}$ groups, which were 99.90% and 99.95%, respectively. Notably, a significant difference ($p < 0.5$) between GOx and $CN_{700}$+vis groups (Fig. 6b, d) was observed, which could be attributed to the enhanced bactericidal performance resulting from the •OH generated by the cascade catalysis. However, the cascade catalytic activity of $CN_{700}$ was relatively low, resulting in a lower bactericidal rate than the Cu/CN+vis experimental group. This result was supported by the in situ EPR detection of the •OH activity produced by $CN_{700}$ and Cu/CN photocatalysis (Fig. S12). Furthermore, SEM images (Fig. 6e) provided additional evidence, showing that MRSA and ESBL *E. coli* in the control group maintained their original plump structures with intact outer membranes, while treatment with GOx caused some structural distortion due to oxidative damage from $H_2O_2$[61]. By contrast, bacteria treated with $CN_{700}$+vis and Cu/CN+vis exhibited significant morphological deformations, such as collapse, distortion, and breakage (highlighted by red arrows), indicating the highest bactericidal efficacy. Additionally, we also investigated the anti-bacterial biofilm capability of Cu/CN under Xe light ($\lambda > 420$ nm) illumination. As shown in Fig. 6f, confocal laser scanning microscopy (CLSM) images of ESBL *E. coli* and MRSA biofilms showed a red fluorescence signal, confirming that Cu/CN under Xe light ($\lambda > 420$ nm) illumination exhibits excellent antibiofilm performance. Therefore, the Cu/CN photocatalytic

therapeutic platform, which utilized a photoswitch to control the generation of ROS to combat MDR bacteria, is highly effective. This platform operated through: i) photocatalytic cascade reaction that consumes glucose to produce •OH, ii) photolysis of water to generate •OH, and iii) oxidation of $O_2$ to form $•O_2^-$ (Fig. 6g).

## Hemocompatibility and cytocompatibility investigation of the Cu/CN

Low cytotoxicity was essential for applying nanomaterials to skin wounds[62]. In vitro cytotoxicity evaluation using NIH/3T3 murine fibroblasts demonstrated that cells cocultured with $CN_{700}$ and Cu/CN (500 μg/mL) showed healthy cellular growth and excellent cell viability with no significant difference versus control group ($p > 0.05$) after 24 h incubation (Fig. 7a, b). Notably, Cu/CN showed no significant cytotoxicity even at elevated concentrations up to 1000 μg/mL (Fig. S26). These results suggested that Cu/CN is well-suited for treating chronic diabetic wounds due to its high biocompatibility. The hemolysis assay evaluated the effect of CN and Cu/CN on red blood cells (RBCs). As seen in Fig. 7c, Triton X-100 acted as the positive control, displaying red color due to complete hemolysis. In contrast, RBCs treated with GOx, CN, and Cu/CN appeared pale yellow and transparent, similar to Tris·HCl, the negative control. Hemolysis ratios of GOx, CN, and Cu/CN were 1.5%, 2.8%, and 3.3%, respectively, all below the permissible limit

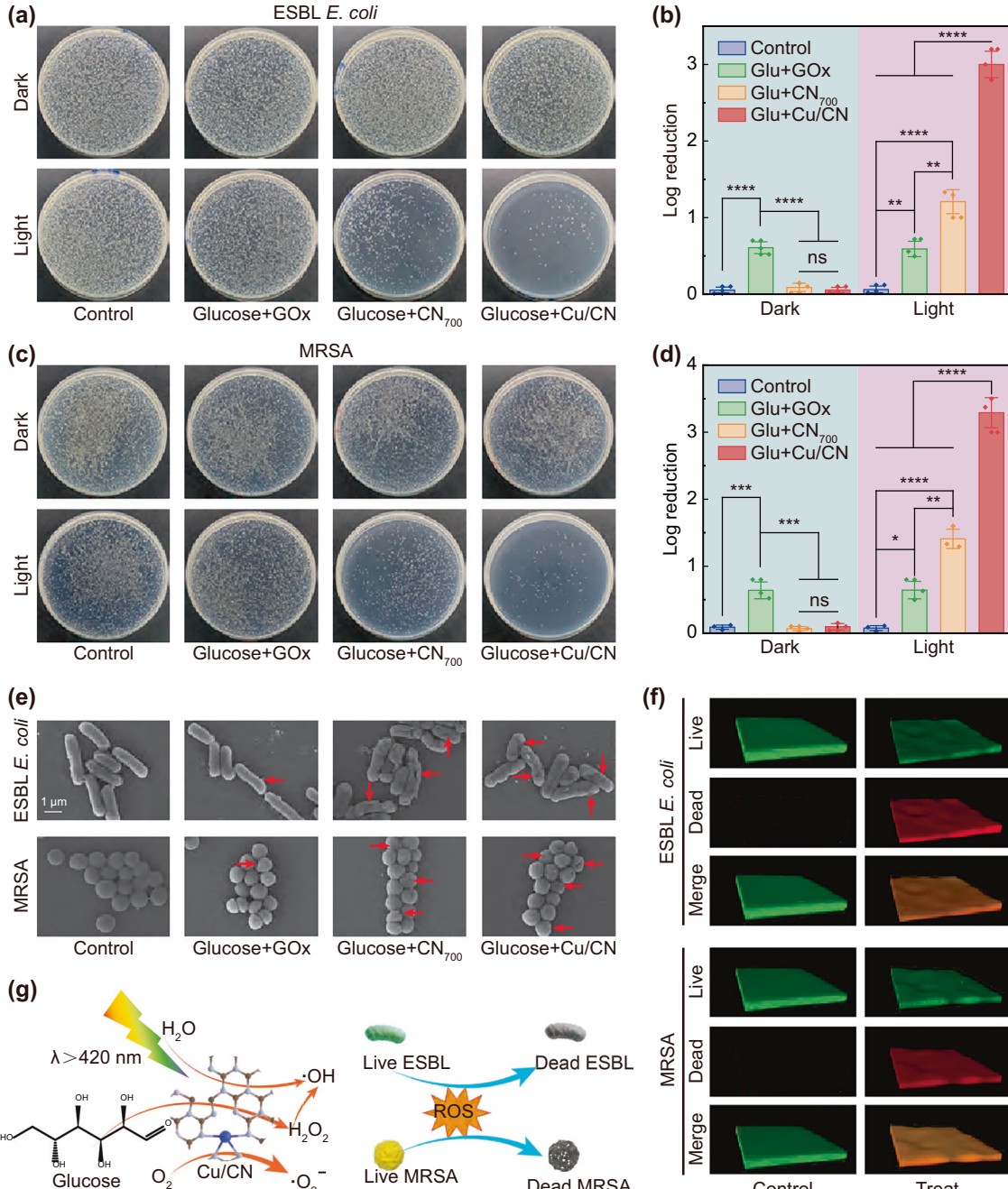

**Fig. 6 | Evaluation of antibacterial capacity.** Photographs of bacterial colonies formed by **a** ESBL *E. coli* and **c** MRSA after treatments with Cu/CN. The relative bacterial viability (*n* = 3 independent experiments) of **b** ESBL *E. coli* and **d** MRSA. The results were expressed as the mean of at least five replicates ±SD (standard deviation). Morphologies of **e** ESBL *E. coli* and MRSA treated or untreated with the Cu/CN group. **f** CLSM photos of biofilm treated with Cu/CN. **g** Schematic diagram of photocatalytic antibacterial control by photoswitch. *P* values were calculated by the one-way ANOVA method. **b** ****(*p* < 0.0001): GOx vs. others for dark, ns (*p* > 0.05): *p* = 0.7161, CN$_{700}$ vs. Cu/CN for dark; ****(*p* < 0.0001): Cu/CN vs. others

and CN$_{700}$ vs control for light, **(*p* < 0.01): p = 0.0099, GOx vs. control for light, **(*p* < 0.01): *p* = 0.0044, GOx vs. CN$_{700}$ for light. **d** ***(*p* < 0.001): *p* = 0.0002, GOx vs. control for dark, ***(*p* < 0.001): *p* = 0.0002, GOx vs. Cu/CN for dark, ns (*p* > 0.05): *p* = 0.9707, CN$_{700}$ vs. Cu/CN for dark; ****(*p* < 0.0001): Cu/CN vs. others and CN$_{700}$ vs. control for light, *(*p* < 0.05): *p* = 0.0164, GOx vs. control for light, **(*p* < 0.01): *p* = 0.0028, GOx vs. CN$_{700}$ for light. Data were presented as mean value ± SD. The experiments for (**b**, **d**) were repeated three times independently with similar results.

of 5%. This indicates that Cu/CN has good hemocompatibility and is suitable for treating diabetic wounds. The cell scratch test further examined the influence of CN and Cu/CN on cell migration. As shown in Fig. 7d, the cell migration area after 48 hours of Cu/CN treatment was larger than in the other three groups, with the migration rates of 68.0%, 45.8%, 75.4%, and 90.4% respectively (Fig. 7e). Cells treated with Cu/CN showed the highest cell migration rate, likely due to trace

amounts of Cu promoting cell migration[63]. Summarily, the excellent cell safety, anti-hemolytic and pro-migration effects of Cu/CN are highly favorable for diabetic wound healing.

**In vivo wound healing performance of Cu/CN**
A diabetic model was established in *Kunming* mice through strepto-zotocin induction, animals exhibiting sustained hyperglycemia (blood

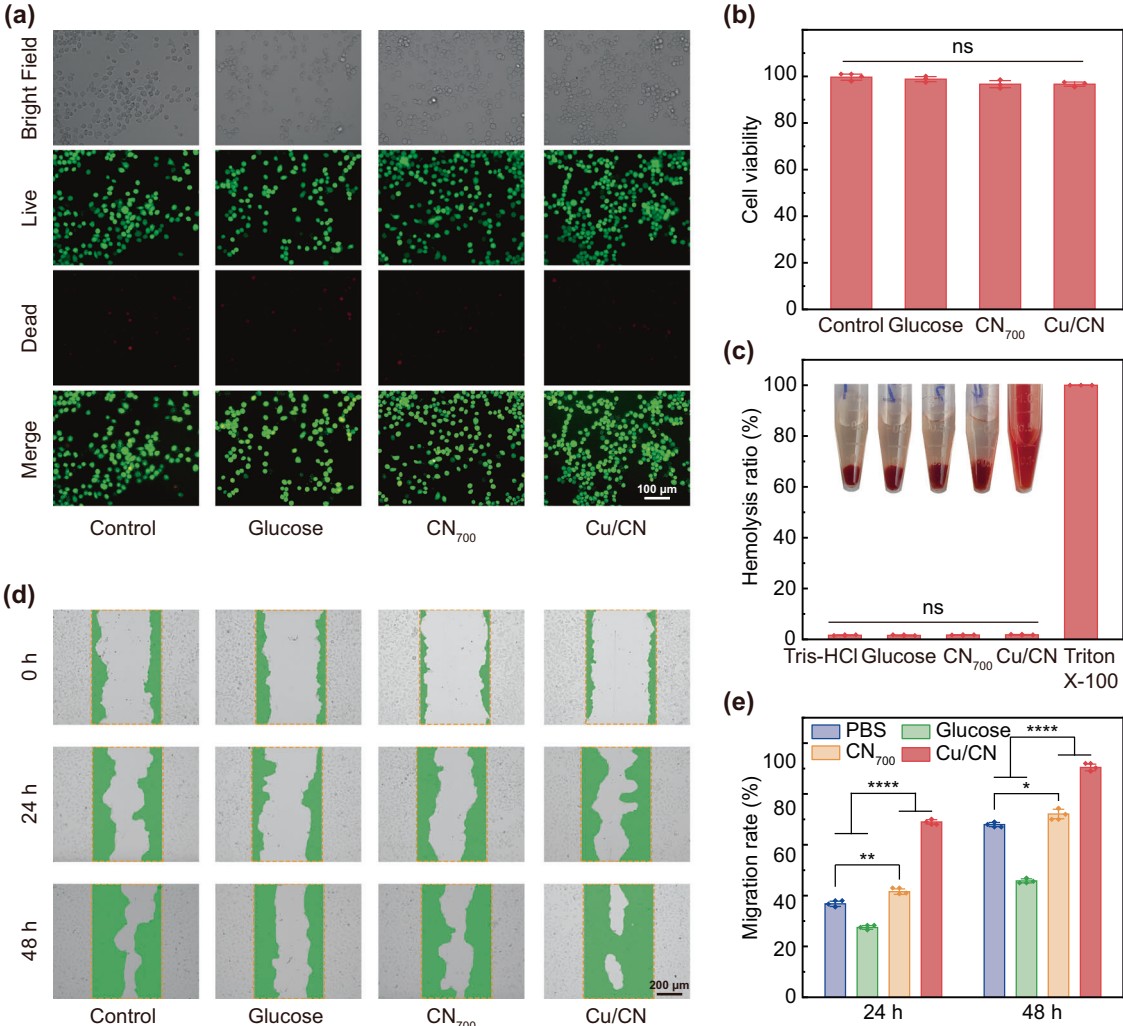

**Fig. 7 | In vitro biocompatible evaluation of Cu/CN. a** Fluorescence images of 3T3 fibroblast cells cultured with PBS, glucose, $CN_{700}$ and Cu/CN after 24 h. **b** Cell viability ($n = 3$ independent experiments) of the $CN_{700}$ and Cu/CN. **c** Hemolysis ratio ($n = 3$ independent experiments) of the $CN_{700}$ and Cu/CN. **d** Cell migration of HUVEC cells in the control and the different groups at 0, 24, and 48 h. **e** The cellular migration after various treatments for different time periods was identified by drawing the green shadow at the edge of cells in (**d**), and calculate the area of cell migration ($n = 3$ independent experiments). *P* values were calculated by the one-way ANOVA method. **b** ns ($p > 0.05$): $p = 0.1194$, Cu/CN vs. control. **d** ns ($p > 0.05$): $p = 0.4922$, Cu/CN vs. Tris-HCl. **e** ****($p < 0.0001$): Cu/CN vs. PBS for 24 h, **($p < 0.01$): $p = 0.0028$, $CN_{700}$ vs. PBS for 24 h; ****($p < 0.0001$): Cu/CN vs. PBS for 48 h, **($p < 0.05$): $p = 0.0356$, $CN_{700}$ vs. PBS for 48 h. Data were presented as mean value ± SD. The experiments for (**b, c, e**) were repeated three times independently with similar results.

glucose levels >16.7 mM) for seven consecutive days were selected for subsequent wound healing studies[14,64]. Full-thickness excisional wounds (6 mm diameter) were created on the dorsal skin using a sterile biopsy punch, then MRSA were inoculated to establish a drug-resistant bacteria-infected diabetic wound model. The $n = 40$ mice were randomly allocated into 4 distinct treatment groups, each with $n = 10$ mice: control, GOx, $CN_{700}$, and Cu/CN. $n = 5$ mice in each group were treated with or without Xe light illumination ($\lambda > 420$ nm, 10 min). Photographs illustrating wound repair across treatment group were shown in Fig. 8a, and quantitative analyses of wound area reduction in diabetic wounds were provided in Fig. 8b. Under light illumination, wound closure ratios at day 14 demonstrated significant therapeutic enhancement: 37.2 ± 2.9% (control), 74.5 ± 0.6% (GOx), 83.6 ± 0.9% ($CN_{700}$), and 97.2 ± 0.4% (Cu/CN) (Figs. S27 and 8e). Notably, Cu/CN-treated wounds achieved near-complete epithelialization, with <3% residual wound area, indicating its superior phototherapeutic efficacy. In contrast, wound healing was significantly delayed in dark conditions, with only 37.3 ± 0.8% closure achieved for Cu/CN-treated group at day 14 (Fig. S28), demonstrating its therapeutic efficacy is light-

dependent. On day 14, wound tissue was collected and homogenized for bacterial number counting. As shown in Fig. 8c, the control group exhibited a high concentration of MRSA, which impeded the wound healing process, whereas the Cu/CN group experienced a substantial reduction in bacterial counts. Notably, all MRSA bacteria on the skin treated with the Cu/CN group were eradicated, underscoring Cu/CN's efficacy in enhancing wound healing and providing photocatalytic antibacterial action in vivo. Cu/CN exhibited effective antibacterial activity upon exposure to Xe light ($\lambda > 420$ nm) illumination, while its nitrogen vacancy-rich CN component photocatalytically consumed glucose, generating $H_2O_2$ through to normalize the glucose levels in diabetic wounds. Cu single-atom further catalyzed $H_2O_2$ produce •OH for antibacterial purposes. The enhanced diabetic wound healing observed with Cu/CN treatment highlights its potential for combining hypoglycemic and antibacterial properties in MDR-infected diabetic wound healing applications.

The transition from the inflammatory to the proliferative stage was a key regulatory point in wound healing. Persistent inflammation was a prominent feature of chronic diabetic wounds[65]. On day 14 of

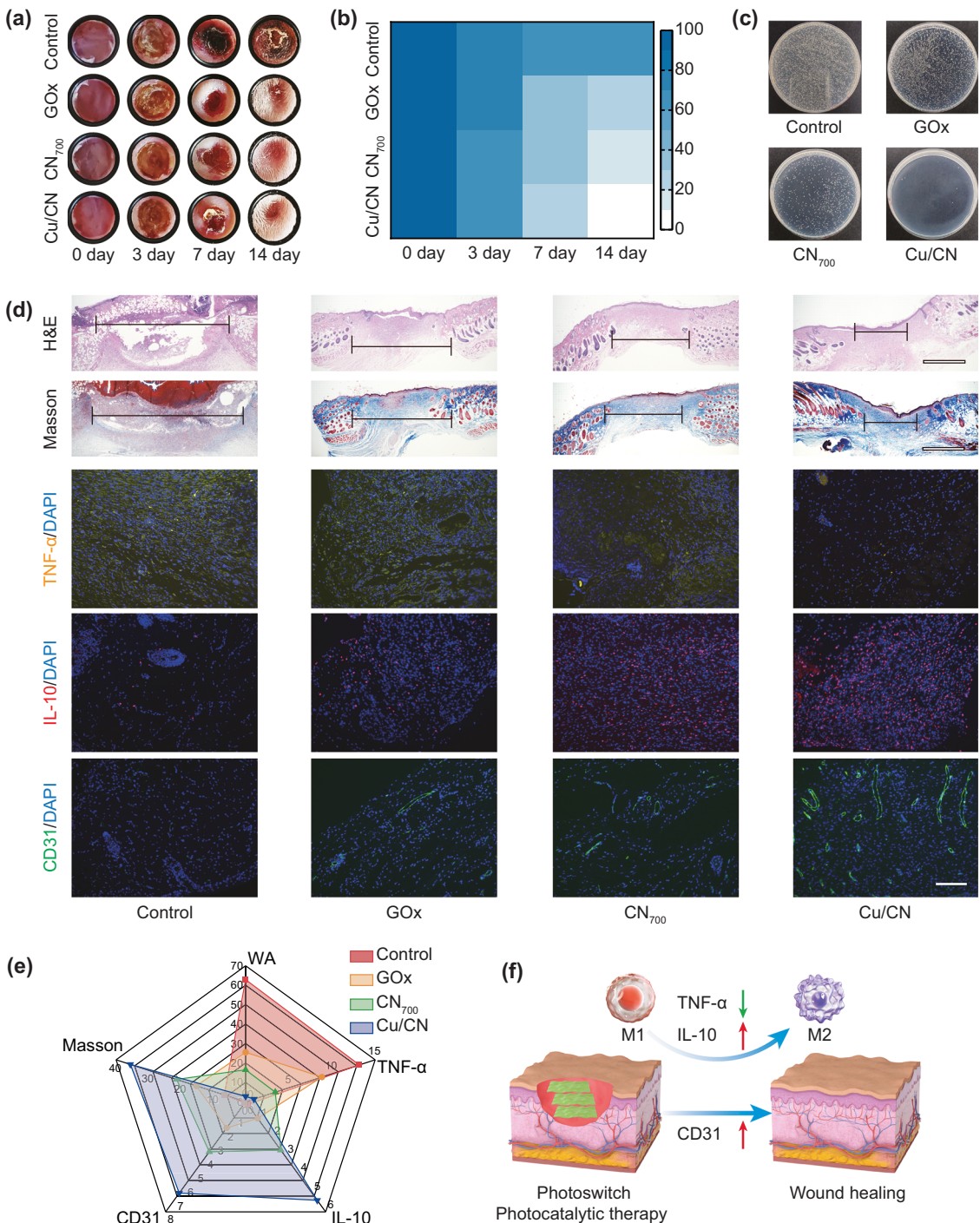

**Fig. 8 | Evaluation of wound healing in diabetic mouse models. a** Photographs of the wounds of wound closure treat with light ($\lambda > 420$ nm). **b** Wound size ratios with different treatments at days 0, 3, 7, and 14. **c** Typical agar plate photos for the remaining MRSA colonies in the ulcer after different treatments on day 14. **d** Immunohistochemical studies of diabetic wounds collected at day 14. Scale bar: 1000 µm for H&E and Masson, 100 µm for TNF-α, IL-10 and CD31. **e** Overall performance of control, GOx, CN, and Cu/CN in the in vivo diabetic wound treatment. **f** Schematic diagram of photocatalytic therapy control by photoswitch. The experiments for (**c, d**) were repeated three times independently with similar results.

treatment, histological analyses were performed. H&E and Masson staining of the wound tissue were utilized to examine the inflammatory reaction and evaluate collagen formation and deposition in wounds treated with control, GOx, CN, and Cu/CN, while immunofluorescence staining for cluster of differentiation 31 (CD31), tumor necrosis factor-alpha (TNF-α), and interleukin-10 (IL-10) was employed to assess angiogenesis and inflammation levels in the healed wounds of these treatment groups (Fig. 8d). In the Cu/CN treated group, complete wound healing was achieved by day 14, with numerous fibroblasts visible under the skin (Fig. 8d). In contrast, wounds in the other groups retained a considerable number of inflammatory cells and showed incomplete healing. CD31 is commonly used to detect the presence of neovascularization in granulation tissue. As depicted in Fig. 8d, the Cu/CN-treated group exhibited the highest number of vascular growths

on day 14 compared to the other three groups, as indicated by CD31 immunohistochemical staining, with CD31 positivity reaching 6.6% (Fig. 8e and Fig. S29). TNF-α, an inflammatory cytokine, can inhibit wound healing in diabetic conditions, while the anti-inflammatory M2 phenotype was usually activated by the Th2 cytokines of IL-10[66]. TNF-α exacerbated inflammation by initiating inflammatory cascades, whereas IL-10 acts as an anti-inflammatory cytokine by suppressing effector T cell activity[67]. Immunohistochemical staining (Fig. 8d) revealed that Cu/CN effectively decreased TNF-α expression while increasing IL-10 expression in inflammatory cells. On day 14, the positive proportion of TNF-α decreased to 1.0%, while IL-10 positivity increased to 5.3% (Fig. 8e and Fig. S29). This controlled treatment process was achieved by using a photocatalytic light switch, allowing ROS generation to be halted after illumination, thus avoiding increased inflammation. Furthermore, after phototherapy completed, Cu/CN could scavenge excess ROS under dark conditions. These findings suggested that Cu/CN promotes the polarization of macrophages from the pro-inflammatory M1 to the anti-inflammatory M2 phenotype and modulates inflammatory cytokine expression (Fig. 8f). This modulation led to anti-inflammatory effects, enhancing the capacity for skin repair in diabetic wounds. To evaluate the biosafety of Cu/CN, we conducted comprehensive toxicity assessments by analyzing serum biochemical markers-including alanine aminotransferase (ALT), aspartate aminotransferase (AST), creatinine (CRE), and blood urea nitrogen (BUN)-as well as histopathological examinations of major organs (heart, liver, spleen, lung, and kidney) following 14 days treatment in mice. Biochemical results demonstrated that 2 weeks Cu/CN treatment elicited no statistically significant alterations in mice hematological parameters and hepatic/renal function indices (Fig. S30). Histopathological analysis also revealed there was no evidence of structural damage or pathological abnormalities in vital organs (Fig. S31). These collective findings confirmed that Cu/CN possesses a favorable in vivo biosafety profile for therapeutic applications.

## Discussion

The Cu/CN photocatalyst presents a multifaceted therapeutic platform with distinct advantages for biomedical applications. The Cu/CN is synthesized through facile thermal polycondensation and impregnation processes, using cost-effective commercially available precursors. The material's properties can be precisely tuned through defect engineering, enabling customized electronic environment for optimized performance. The visible light-responsive nature of this photocatalyst renders it particularly suitable for superficial tissue applications, as the penetration depth of these wavelengths optimally matches dermal thicknesses, making it ideal for managing cutaneous pathologies such as diabetic wounds. This system demonstrates multiple therapeutic efficacies by simultaneously addressing localized hyperglycemia, providing robust antibacterial activity, and regulating inflammation, which synergistically enhance diabetic wound repair processes. Furthermore, as a functional nanomaterial, Cu/CN exhibits significant potential for integration into advanced delivery systems such as microneedle patches or nanovectors, to expand its clinical applicability.

In conclusion, we have successfully developed a Cu/CN bionanomaterial that can effectively treat MDR bacteria-infected diabetic wounds by simultaneously regulating blood glucose level, bacterial infection, and persistent inflammation. The synergistic effects of N vacancy and single-atom in Cu/CN enable the realization of a photoswitchable cascade reaction. The introduction of N vacancies enhances the activity of photocatalytic oxidation for glucose consumption, which helps to reduce blood glucose level at the wound site. Subsequently, the $H_2O_2$ generated in the previous step is converted into •OH through a Cu single-atom photocatalytic cascade reaction. Concurrently, Cu/CN can directly produce •OH and •$O_2^-$ through

photocatalytic water splitting. DFT calculation confirmed that glucose preferentially adsorbs at N vacancy, while $H_2O_2$ adsorbs more readily at Cu single-atom. In vitro antibacterial results demonstrate that the Cu/CN photocatalyst has significant bactericidal effects on drug-resistant MRSA and ESBL *E. coli* and exhibits anti-biofilm activity. The in vivo study in a mice diabetic wound MRSA infection model shows that Cu/CN photocatalyst has good biological safety and excellent therapeutic efficacy with visible light illumination. The excess ROS are scavenged by Cu/CN under dark conditions, reducing oxidative stress and promoting M1 to M2 macrophage polarization to resolve inflammation and enhance wound healing. This work presents a proof of concept for using a photoswitchable cascade reaction to treat MDR bacterial-infected diabetic wound complications.

## Methods

### Materials

Melamine (≥99%, M108433), glucose oxidase (GOx, EnzymoPure, >180 U/mg, G109029), cupric nitrate trihydrate (≥99%, C140879) and glucose (AR, G116300) were purchased from Aladdin Industrial Co., Ltd. (Shanghai, China). The rest reagents were obtained from Sinopharm Chemical Reagent Co., Ltd. (Shanghai, China).

### Preparation of CN

3 g of melamine was put into a porcelain cup with a cap, then calcined at 550 °C for 4 h with a ramping rate of 2.2 °C min$^{-1}$ in a muffle furnace in air. After heating, the resulting yellow agglomerates (about 1.5 g) were gently ground and treated under ultrasonication for 3 h as an aqueous solution (1 g L$^{-1}$). Then the powder was filtered through a 0.45 μm membrane, washed three times by deionized water, and dried at 80 °C for further tests (the sample was named as $CN_{550}$).

### Preparation of nitrogen vacancy-rich CN

For the quick thermal treatment, a horizontal tube furnace was heated to target temperatures (600 °C, 650 °C, 700 °C and 750 °C) and 250 mg of $CN_{550}$ was put in an alumina crucible. A stopwatch was started as soon as the alumina crucible was placed in the targeted heating zone. When the watch ran for 5 min, the crucible was fished out as soon as possible and cooled down in the air. The whole process is carried out in well-ventilated space. The resulting products were collected and named as $CN_{600}$, $CN_{650}$, $CN_{700}$, $CN_{750}$, according to the target temperature. When the treatment temperature was up to 800 °C, the $CN_{550}$ was completely decomposed.

### Synthesis of Cu/CN

1 g $CN_{700}$ was dispersed into 10 mL 5 wt% cupric nitrate trihydrate solution, stirred at room temperature for 1 h, and evaporated at 80 °C. The yellow-green powder was washed with deionized water and dried at 80 °C overnight. 5 mg of catalyst was dissolved in 5 mL HCl (12 mol/L) with continuous stirring for complete digestion. The digestion mixture underwent 100× dilution using deionized water, followed by volumetric standardization in a 500 mL capacity flask and measured with ICP-OES (Agilent 5110). Through ICP analysis, the actual loading amounts were 1 wt%, abbreviated as Cu/CN.

### Characterization

The morphology and size of CN were measured by TEM (HT7800, Hitachi, and JEM-200F). X-ray diffraction (XRD) patterns were recorded on a D8 Advance diffractometer (Bruker, Germany, Cu Kα, $\lambda = 1.54056$ Å) operated at 40 kV and 200 mA at room temperature. The X-ray photoelectron spectroscopy (XPS) was measured on a Thermo Scientific Kα photoelectron spectroscopy with Al $K_\alpha$ radiation. The surface morphology and element mapping of CN and Cu/CN were performed using a field emission scanning electron microscope (FE-SEM, ZEISS Gemini SEM 300) after lyophilization. Aberration-corrected high-angle annular dark-field scanning transmission

electron microscopy (AC-HAADF-STEM) characterization was conducted on an FEI Themis Z. The chemical groups of specimens were analyzed employing a Fourier transform infrared (FTIR) spectrometer (Tensor II, Bruker, Germany). Raman spectra were collected by Renishaw inVia Raman Microscope using the 785 nm laser as the excitation source. The diffuse reflectance UV–vis (DR UV–vis) spectra were recorded on the UV–vis spectrophotometer (UV-3600 Plus, Shimadzu). The electron paramagnetic resonance (EPR) spectra were tested via a Bruker EMXPLUS10/12 EPR electron paramagnetic resonance spectrometer.

### Elemental analysis of CN
Elemental analysis of CN was conducted on a UNICUBE analyzer (Elementar, Germany). Samples (5.0 mg) were combusted at 950 °C with sulfanilamide as the standard. The weight percentages (wt%) of sulfanilamide elements are as follows ($C = 41.84\%$, $H = 4.68\%$, $N = 16.23\%$, $S = 18.63\%$) ($n = 3$ independent experiments).

### Hypoglycemic ability of CN and Cu/CN
5 mg CN with different nitrogen vacancies were dispersed in the 10 mL glucose solution (20 mmol/L) and irradiated using an Xe lamp cutoff filter 420 ($\lambda > 420$, $0.5$ W/cm$^2$, 10 min), and the consumption of glucose is measured by the amount of $H_2O_2$ produced. 5 mg Cu/CN was dispersed in 10 mL glucose solution (20 mmol/L), irradiated with Xe lamp cut-420 ($\lambda > 420$ nm, $0.5$ W/cm$^2$, 10 min), and measured glucose consumption using a glucose assay kit (Solarbio, BC2505).

### ROS-scavenging ability evaluation
The antioxidant capacity of Cu/CN was evaluated by the DPPH radical scavenging ratio (%). 5 mg of Cu/CN are mixed with DPPH or ABTS ethanol solution (100 μM, 10 mL, different pH was adjusted using formic acid), respectively. After incubation at 37 °C in the dark for 6 h, the absorbance of the solution was measured at 517 or 734 nm by UV-vis spectrophotometer ($n = 3$ independent experiments).

### Products analysis of photocatalytic glucose consumption
The sample was tested on a ThermoFisher ICS5000 HPLC system using a Dionex Carbopac$^{TM}$ PA10 ($4 \times 250$ mm) column and electrochemical detector to separate and detect the main products. The mobile phase was $H_2O$ (A), 500 mM NaOH and 50 mM NaAc (B) and 20 mM NaOH (C) with flow rate of 1.0 mL/min. The separation was performed at 30 °C and the injection volume was 25 μL.

Elution process:

| Time(min) | A% | B% | C% |
|---|---|---|---|
| 0.000 | 0.0 | 0.0 | 100.0 |
| 30.000 | 50.0 | 0.0 | 50.0 |
| 30.100 | 70.0 | 30.0 | 0.0 |
| 46.000 | 70.0 | 30.0 | 0.0 |

### Calculation of the charge carrier density (Nd)
As shown in Fig. 3e, the positive slope of the M-S curve indicates that $CN_{700}$ are n-type semiconductors and thus, their $N_d$ can be calculated by the following equation

$$N_d = 2/(e \times \varepsilon_0 \times \varepsilon) \times (d(1/C^2)/dVs))^{-1} \qquad (1)$$

where C is the capacitance of the space charge layer; $\varepsilon_0$ denotes the vacuum dielectric constant ($8.85 \times 10^{-14}$ F/cm); $\varepsilon$ is the dielectric constant of the sample (15.6 for g-$C_3N_4$); e is the electron charge ($1.602 \times 10^{-19}$ C).

### Calculation of the standard hydrogen electrode
The potential of the catalyst, relative to the standard hydrogen electrode, can be calculated from the Nernst equation

$$E_{RHE} = E_{Ag/AgCl} + 0.059 \times pH + E^{\circ}_{Ag/AgCl} \qquad (2)$$

### In vitro antibacterial performance of CN and Cu/CN
The methicillin-resistant *Staphylococcus aureus* (MRSA, Gram-positive, ATCC 43300) and extended-spectrum $\beta$-lactamases producing *Escherichia coli* (ESBL *E. coli*, Gram-negative, Xi'an Jiaotong University) were used to evaluate the antibacterial performance of all samples. Briefly, stock bacteria were added into 5 mL of sterilized fresh LB liquid medium and cultivated at 37 °C for 4–6 h. Then, the concentration of bacterial suspensions was measured by utilizing a multifunctional microplate reader (OD600) at 600 nm. The contact-active antimicrobial efficacy of $CN_{700}$ and Cu/CN was determined using a reported protocol. Typically, 20 μL of $1 \times 10^6$ colony-forming units mL$^{-1}$ (CFU/mL) bacterial solution was mixed with 20 μl Cu/CN solution (500 μg/mL), and 20 μL of the bacterial solution alone was used as control. 180 μL of aseptic PBS was added to suspend residual bacteria after being incubated with or without light ($\lambda > 420$ nm) for 10 min. 50 μL resuspension was dropped onto a standard agar plate and then cultured for 12 h at 37 °C to observe the surviving bacterial colony. To count the surviving bacteria colony, various 10-fold gradient dilutions in aseptic PBS were spread onto agar plates and the survivors were obtained through images of plates after cultured. Each group was repeated three times, and the killing rate was calculated by the following equation

$$\log \text{reduction} = \log (\text{cell count of control}/ \text{survivor count treated with materials}) \qquad (3)$$

### In vitro antibiofilm tests
100 μL MRSA/ESBL *E. coli* solution ($1 \times 10^6$ CFU/mL) was added to a dish containing LB solid medium and incubated at 37 °C for 24 h to form complete biofilms. Cu/CN was dispersed in 20 mM glucose solution to a final concentration of 500 μg/mL. The biofilms were treated with 100 μL the above dispersion solution for 10 min under light ($0.5$ W/cm$^2$) or dark conditions. The supernatant was then removed and the biofilm at the bottom was washed three times with PBS (pH 7.4). Four groups were stained with Syto-9 and PI for 30 min at room temperature and photographed by CLSM to obtain three-dimensional fluorescence images.

### In vitro hemolysis assay
The hemolysis test was performed using fresh whole blood from *Kunming* mice. First, 1 mL of fresh mice blood was centrifuged for 10 min at $400 \times g$ using a centrifuge to collect red blood cells (RBC) and washed with Tris-HCl 3 times. Then 500 μL of RBC Tris-HCl solution (5%) was added to 500 μL of Tris-HCl solution containing different materials, in which Tris-HCl and Triton X-100 was used as negative and positive controls, respectively. The mixture was centrifuged at $400 \times g$ for 10 min after being incubated at 37 °C for 1 h centrifuge it to test the absorbance of the supernatant at 540 nm. The formula for calculating hemolysis rate (%) is: hemolysis rate (%) = $(A_M - A_N)/(A_W - A_N) \times 100\%$, where $A_M$, $A_N$ and $A_W$ represent absorbance after co-incubation of red blood cells with the materials, Tris-HCl and Triton X-100, respectively.

### In vitro cytotoxicity assay
The $CN_{700}$ and Cu/CN (500 μg/mL, 10 μL) were mixed with NIH/3T3 murine fibroblasts (STCC20005P, Wuhan Servicebio Technology Co., Ltd.) was added to a 96-well microtiter plate and incubated at 37 °C for 24 h. The growth and proliferation of 3T3 fibroblasts with the $CN_{700}$

and Cu/CN were observed using an optical microscope. Then the cells were stained with Almar Blue and incubated at 37 °C for 4 h. The fluorescence intensity was calculated with a microplate reader to quantify cell activity. Each test was repeated three times.

### Diabetic chronic wound healing experiment

*Kunming* male mice (6 weeks, outbred, 30–40 g) were provided by Beijing Vital River Laboratory Animal Technology Co., Ltd. and housed in groups (n = 5 mice) in plastic cages on a 12-h light/dark cycle. The temperature and humidity were kept at 24–26 °C and 60%, respectively. All animal-based experimental studies were conducted in accordance with the guidelines of the Administration of Laboratory Animals of China and approved by the animal ethics committee of Northwestern Polytechnical University (Number: NPU201906). The wound healing performance was studied using a murine diabetic wound healing model. A single injection of STZ (150 mg/kg) was used to establish the diabetic mouse model, and 20 μL of masr bacterial solution ($1 \times 10^6$ CFU/mL) was added to the wound to construct an infection model, then these mice were divided into eight different treatment groups, randomly, five in each group: Control+vis, Control−vis, GOx+vis, GOx−vis, $CN_{700}$+vis, $CN_{700}$−vis, Cu/CN+vis and Cu/CN−vis. The diabetic wounds in the respective groups were dripped with 100 μL $CN_{700}$ or Cu/CN PBS solution (500 μg/mL), then irradiated with Xe lamp cut-420 ($0.5 \, W/cm^2$) for 10 min on the first day. The mice were killed by anesthesia on days 3, 7 and 14 days, and the wound tissue was collected for later tests. Hematoxylin and eosin (H&E), Masson, and CD31 staining were used for the histological analysis. The unrepaired area ratio of the wound was calculated by equation:

$$\text{Wound area ratio}(\%) = ((A_x - A_y)/A_y) \times 100\% \quad (4)$$

where $A_x$ and $A_y$ are the wound area of the infected and primary area.

### Density functional theory calculation

First-principles calculations were performed by using periodic DFT with the Perdew–Burke–Ernzerhof (PBE) exchange-correction functional under the generalized gradient approximation. The plane waves forming the wave functions were expanded with an energy cutoff set to 500 eV. A gamma-centered k-point mesh of $3 \times 3 \times 1$ was applied for the geometry optimization. The convergence criteria for the geometry optimization were established at $1.0 \times 10^{-6}$ eV/atom for the total energy and 0.02 eV/Å for the force. To minimize the interaction between surfaces in the periodically repeated slabs, a substantial vacuum gap of 15 Å was introduced. The energy of adsorption was determined by using a standard formula, as follows:

$$E_{ads} = E_{Total} - E_{Cat} - E_{molecular} \quad (5)$$

In the calculations of free energies, adjustments for entropic corrections and zero-point energy (*ZPE*) were incorporated. The free energy of various species was determined based on the following standard formula:

$$\Delta G = E + \Delta ZPE + \Delta H - \Delta TS \quad (6)$$

where *ZPE* represents the zero-point energy, $\Delta H$ is the change in enthalpy or the integrated heat capacity, *T* denotes the temperature at which the system or product is analyzed, and *S* signifies the entropy of the system.

### In vitro ROS scavenging activities of Cu/CN

Initially, RAW 264.7 murine macrophages (STCC20020G, Wuhan Servicebio Technology Co., Ltd.) were seeded into cell culture plates and stimulated with $H_2O_2$ (0.5 mmol/L) to produce ROS. The cells were then co-incubated with PBS, $CN_{700}$, and Cu/CN (500 μg/mL). The intracellular ROS levels were subsequently measured by fluorescence microscopy, following staining with the ROS-specific probe, 2,7-dichlorodihydrofluorescein diacetate (DCFH-DA).

### In vitro cell migration assay of Cu/CN

Human umbilical vein endothelial cells (HUVECs, CP-H082, Procell Life Science & Technology Co., Ltd.) were seeded in 6-well plates at a density of $5 \times 10^5$ cells per well and cultured until 90–100% confluent monolayers formed within 24 h. Uniform scratch wounds were created using a sterile p200 pipette tip, followed by three gentle PBS washes to remove detached cells. Following medium exchange to serum-free DMEM conditions, $CN_{700}$ and Cu/CN were independently introduced into the culture system and co-incubated with the cells in separate wells. Photographs were taken with a microscope (EVOS FL Auto 2, Thermo Fisher Scientific) at 0, 24, and 48 h, respectively. Migration rates were quantified by calculating the percentage of wound closure relative to the initial wound area using ImageJ software.

### Statistical analysis

Statistical analysis was performed using GraphPad Prism 8.0 software (GraphPad Inc., San Diego, CA, USA). Quantitative data were analyzed using the one-way ANOVA test. $P < 0.05$ was considered to be a statistically significant difference. Experimental data were obtained from at least three independent replicates.

### Reporting summary

Further information on research design is available in the Nature Portfolio Reporting Summary linked to this article.

## Data availability

All data underlying this study are available from the corresponding author upon request. Source data of Figs. 2a–f, 3a–e, 4a–e, 6b, d, 7b, c, e, and 8e, and Supplementary Figs. S3-S5, S7-S10, S12-S17, S19, S22, and S26-S30 are provided in a Source Data file. Source data are provided with this paper.

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

## Acknowledgements

This research was sponsored by the key research program of Ningbo (2023Z210 to Y.Z.), the National Foreign Expert Project (H20240307 to Y.Z.), the National Natural Science Foundation of China (52473265 to P.L.), and the Shaanxi Provincial Science Fund for Distinguished Young Scholars (2023-JC-JQ-32 to P.L.).

## Author contributions

X.S. and P.Z. conceived the idea and designed the experiments; P.L. and Y.Z. supported the characterizations and acquired the funding; X.S. and P.Z. conducted the experiments with the assistance from L.T., P.W., N.L., and Q.W.; Y.-R.L. conducted the Calculation; X.S. and P.Z. analyzed the data and interpreted the results. X.S., P.L., and Y.Z. wrote and revise the manuscript. All authors read, discussed, and commented on the manuscript.

## Competing interests

The authors declare no competing interests.
