## [Transparent Peer Review file · Nature Communications]

Cu Single-Atom Embedded g-C₃N₄ nanosheets Rehabilitate Multidrug-Resistant Bacteria Infected Diabetic Wounds via Photoswitchable Cascade Reaction

Corresponding Author: Professor Peng Li

Version 0:

Reviewer comments:

Reviewer #1

(Remarks to the Author)

The manuscript presents a novel therapeutic strategy utilizing Cu single-atom embedded g-C₃N₄ nanosheets (Cu/CN) for the treatment of multidrug-resistant (MDR) bacterial-infected diabetic wounds. The approach leverages a photoswitchable cascade reaction to regulate glucose levels, antibacterial activity, and inflammation. The study is well-designed, and the results are promising, demonstrating significant antibacterial efficacy and wound healing in diabetic mouse models. However, the catalytic mechanism requires further clarifications, thus, the reviewer suggests reconsideration after addressing the following comments.

1. The manuscript claims that the nitrogen vacancies in g-C₃N₄ enhance photocatalytic glucose oxidation and subsequent conversion to H₂O₂, which is then further converted to hydroxyl radicals (\bullet OH) via a Cu single-atom-mediated cascade reaction. However, the detailed mechanistic pathway of how nitrogen vacancies and Cu single-atoms synergistically enhance the photocatalytic activity is not fully elucidated. Could the authors provide more detailed DFT calculations or other evidence to support the proposed mechanism?

Additionally, the role of Cu single-atoms in the photocatalytic water splitting process to generate \bullet OH and \bullet O₂⁻ needs further clarification. Are there any intermediates or transition states involved in this process?

2. The manuscript claims that \bullet OH and superoxide radicals \bullet O₂⁻ are generated through photocatalytic pathways. However, the evidence supporting this claim is not sufficiently detailed. Given that glucose oxidation to glucuronic acid can lower the microenvironment's pH, which may enhance mimics enzyme activity of Cu/CN for ROS production, it is crucial to provide clear and direct evidence that \bullet OH and \bullet O₂⁻ are indeed produced via photocatalytic mechanisms rather than other pathways.

3. It is necessary to provide the ability of Cu/CN to clear ROS under different pH conditions. Can Cu/CN clear ROS in an acidic environment induced by glucuronic acid?

4. The manuscript reports over 99.9% antibacterial activity and effective biofilm inhibition. However, the photocatalytic efficiency under different light intensities and wavelengths is not thoroughly investigated. Could the authors provide data on the photocatalytic performance under different light conditions (e.g., different wavelengths, intensities) to better understand the practical applicability of this system?

5. The manuscript mentions that Cu/CN scavenges excess ROS in the dark, reducing inflammation. However, the exact mechanism by which Cu/CN scavenges ROS in the absence of light is not clearly explained. Could the authors provide more detailed experimental evidence or mechanistic insights into this process?

Reviewer #2

(Remarks to the Author)

This manuscript "Cu Single-Atom Embedded g-C₃N₄ nanosheets Rehabilitate Multidrug-Resistant Bacteria Infected Diabetic Wounds via Photoswitchable Cascade Reaction" presents the preparation of a rational designed photocatalyst and their application in bacteria infected diabetic wounds therapy through programmed reactions. It is an interesting exploration to use multifunctional single atom catalysts for biomedical applications. This photocatalyst could consume the excess glucose and eliminate bacteria at diabetic wounds under light irradiation, and clear excess ROS without light to reduce inflammation and promote wound healing. In vitro studies demonstrate over 99.9% antibacterial efficacy and strong anti-biofilm ability towards multidrug resistant bacteria strains. In vivo animal studies exhibited good biological safety and excellent therapeutic efficacy. The major findings of this work are intriguing and meaningful, with relatively sufficient

theoretical and experimental exploration. I recommend its acceptance for publication in this journal after minor revision. My comments are listed below:

- 1) The title mentions the use of "g-C3N4 nanosheets", but this term is not mentioned or explained in the main text. It is recommended that the author check their contents.
- 2) Why nitrogen vacancies of g-C3N4 could enhance the oxidation of glucose? The authors should provide more clear explanation and evidence.
- 3) Some important details are missed in their biological study. What is the concentration of Cu/CN used in the cell experiments? What is the concentration of Cu/CN used for in vivo treatment? How many times and how long was Cu/CN used for in vivo treatment experiments?
- 4) It is necessary to supplement the cytotoxicity study of Cu/CN at different concentrations.
- 5) P18, line9, "requires NIR irradiation for effective activity", please verify this content.
- 6) The text contains multiple instances where there is no space before parentheses and inconsistent font usage. It is recommended that the author thoroughly review and revise the entire manuscript to address these article format issues.

Reviewer #3

(Remarks to the Author)

The work present by Sun et al. is novel to some extent, robust and presents significant fundamental and applicative implications and advancements. Most of the conclusions and claims are well supported by data, experiments and analysis. It is also relevant and important to the field of study. The quality of the work satisfies the high standards desired for consideration in Nat. Commun. Thus, I would recommend reconsidering this work for acceptance after the following minor changes and suggestions are addressed.

There are several grammatical and sentence construction errors at various places in the manuscript. Please carefully proofread and rectify those.

The prominent peak in FTIR at 2175 cm^{-1} is not that prominent.. also it appears to be present in a similar fashion in all the annealed samples. I fail to identify a significant change for the different thermally annealed vs. control. Please explain/justify

Please provide elemental ratio (C/N) and Raman peak shifts (particularly corresponding to the triazine rings) to further prove/probe nitrogen vacancies in different thermally etched samples.

Does the N-vacancies lead to a red shift in UV-Vis abs? -- Please check and comment.

Please conduct TRPL measurements to analyze the charge-carrier lifetime for different samples.

The authors didn't mention about the quantitative Cu loading in the Cu/CN sample in the manuscript (only in SI). Please include it in the discussion combining results from ICP/XPS and EDS.

The authors should provide HAADF-STEM images to show the single atom dispersion of Cu in the CN matrix.

The experimental/methodology section of the paper is not elaborate enough—many of the characterization and details are missing. The authors should provide all the information in detail to ensure reproducibility. Many characterization details and specifications are missing (please check carefully and provide the necessary info). How were the materials used in the bandages in the animal model would healing experiments?

Reviewer #4

(Remarks to the Author)

The study presents a promising strategy for diabetic wound management with robust in vitro and in vivo results. Addressing the above points will enhance the manuscript more readable.

- 1、 The title emphasizes "photoswitchable cascade reaction," but the mechanism of light-switching (on/off control) needs clearer experimental validation. The abstract is comprehensive but could benefit from conciseness, focusing on the most impactful findings.
- 2、 The synergistic effect of nitrogen vacancies and Cu single-atom embedding is a key innovation. However, the introduction should more explicitly contrast this approach with prior studies using defect engineering or single atoms alone, emphasizing the novelty of combining both strategies.
- 3、 The synthesis of Cu/CN (e.g., calcination temperatures, Cu loading ratios) requires more precise parameters to ensure reproducibility. Additionally, HAADF-STEM or AC-STEM images are necessary to confirm the atomic dispersion of Cu, as XRD alone cannot rule out nanoparticle formation.
- 4、 The claim that Cu/CN scavenges excess ROS in the dark lacks direct evidence. Quantifying ROS levels (e.g., via fluorescence probes) under dark conditions would strengthen this conclusion.
- 5、 While bactericidal rates are impressive, statistical significance markers (e.g., p-values) are missing in Figure 5b/d. Clarify the sample size and replicate numbers for bacterial viability assays.
- 6、 Hemolysis and cell viability assays are well-presented, but long-term cytotoxicity (e.g., 7-day exposure) and in vivo toxicity (e.g., histopathology of major organs) should be included to address potential safety concerns.
- 7、 The adsorption energy calculations for glucose and H_2O_2 are insightful. However, provide additional computational evidence (e.g., reaction pathways, activation barriers) to support the proposed cascade mechanism.

8、 The discussion should address scalability (e.g., cost of Cu/CN synthesis) and practical challenges (e.g., light penetration depth in human tissue) for translational applications.

9、 Ensure all cited references (e.g., Ref. 6) are peer-reviewed and publicly available.

Version 1:

Reviewer comments:

Reviewer #1

(Remarks to the Author)

1. Although the authors have proved the production of arabinose rather than the glucuronic acid, the byproduct HCOOH is highly acidic that can also significantly reduce the pH value in the micro environment. In general, the POD activity of nanozyme can be activated at low pH value, therefore, it is hard to say the ROS production was triggered by light or the low pH value. The reviewer suggest the author to study the production of ROS (by ESR) under low pH value in the absence of light to clarify this point.

2. In the response section, The author claim that "The reduction reaction of H₂O₂ generated by single electron transfer of O₂ at the conduction band position was equation 3." Equation 3 doesn't seem right. In this reaction the superoxide should be firstly produced with one electron injection from the CB.

3. Can the author provide more details about the TS in Figure 4c.

Reviewer #2

(Remarks to the Author)

The present manuscript had been revised in an appropriate and disciplined manner. The revision increased the quality of the manuscript and hence the manuscript may be suitable for its publication in its present form.

Reviewer #3

(Remarks to the Author)

The authors have satisfactorily addressed all the concerns and queries raised by the reviewers through additional experiments and appropriate revisions. The comprehensive effort in responding to the all the referees' feedback has significantly enhanced the quality and clarity of the manuscript and its claims. I, therefore, recommend the acceptance and publication of the manuscript in its current form.

Reviewer #4

(Remarks to the Author)

After authors' carefully revision and answer, all questions are resolved and the manuscript is ready for publication.

Version 2:

Reviewer comments:

Reviewer #1

(Remarks to the Author)

The authors have addressed my concerns. The manuscript can be accepted for publication in this journal.

Response to the reviewers' comments

Reviewer #1 (Remarks to the Author):

The manuscript presents a novel therapeutic strategy utilizing Cu single-atom embedded g-C₃N₄ nanosheets (Cu/CN) for the treatment of multidrug-resistant (MDR) bacterial-infected diabetic wounds. The approach leverages a photoswitchable cascade reaction to regulate glucose levels, antibacterial activity, and inflammation. The study is well-designed, and the results are promising, demonstrating significant antibacterial efficacy and wound healing in diabetic mouse models. However, the catalytic mechanism requires further clarifications, thus, the reviewer suggests reconsideration after addressing the following comments.

1. The manuscript claims that the nitrogen vacancies in g-C₃N₄ enhance photocatalytic glucose oxidation and subsequent conversion to H₂O₂, which is then further converted to hydroxyl radicals (•OH) via a Cu single-atom-mediated cascade reaction. However, the detailed mechanistic pathway of how nitrogen vacancies and Cu single-atoms synergistically enhance the photocatalytic activity is not fully elucidated. Could the authors provide more detailed DFT calculations or other evidence to support the proposed mechanism?

Additionally, the role of Cu single-atoms in the photocatalytic water splitting process to generate •OH and •O₂⁻ needs further clarification. Are there any intermediates or transition states involved in this process?

Response: Thanks for the reviewer's valuable comment. Following your suggestion, additional experiments and DFT calculations were supplemented to fully elucidate the detailed mechanistic pathways.

The products of photocatalytic glucose consumption were analyzed by HPLC. The determined main product was arabinose, and there was no glucuronic acid found (Figure S21). So the complete equation would be:

The oxidation reaction occurring at the valence band position is

The reduction reaction of H₂O₂ generated by single electron transfer of O₂ at the conduction band position is

DFT calculations were performed for both reactions (2) and (3), with the corresponding energy profiles presented in newly supplemented Figure 4c-d and Figure S22-23. The upstream step is endothermic and non-spontaneous, whereas the downstream step is exothermic and spontaneous. The step with the highest endothermic energy corresponds to the rate-determining step (RDS) of the overall reaction. A lower energy barrier for

the RDS leads to enhanced catalytic activity. The presence of Cu single-atom catalysts significantly reduces the energy barrier of the rate-determining step in reaction (2), thereby facilitating the reaction kinetics. For reaction (3), the Cu single-atom catalyst enables the formation of H₂O₂ with the lowest energy barrier, enhancing the thermodynamic favorability of this pathway. Furthermore, the subsequent generation of •OH radicals proceed with a relatively low activation energy barrier over the Cu single-atom sites, indicating a kinetically favorable process. Therefore, based on the experimental results (**Figure 2a**) and charge density difference analysis (**Figure S19-S20**), the presence of N vacancies facilitates the oxidation of glucose, while the Cu single-atom sites promote the formation of •OH radicals with enhanced reaction kinetics.

The DFT calculation results reveal that the introduction of Cu single-atom significantly reduces the reaction energy barrier in photocatalytic water splitting, thereby enhancing the reaction kinetics. The rate-limiting step involves the cleavage of the O–H bond, which serves as the key intermediate transition state in this process (**Figure 4e** and **S24**). The relevant description and discussion were supplemented in the revised manuscript main text (Page 14) and also shown below:

“The products of photocatalytic glucose consumption were analyzed by HPLC. The determined main product was arabinose, and there was no glucuronic acid found (Figure S21). The complete equation would be:

The oxidation reaction occurring at the valence band position was

The reduction reaction of H₂O₂ generated by single electron transfer of O₂ at the conduction band position was

DFT calculations were performed for both reactions, with the corresponding energy profiles presented in Figure 4c-d and Figure S22-S23. The upstream step was endothermic and non-spontaneous, whereas the downstream step is exothermic and spontaneous. The step with the highest endothermic energy corresponded to the rate-determining step (RDS) of the overall reaction. A lower energy barrier for the RDS leads to enhanced catalytic activity. The presence of Cu single-atom catalysts significantly reduced the energy barrier of the rate-determining step in reaction (2), thereby facilitating the reaction kinetics. For reaction (3), the Cu single-atom catalyst enabled the formation of H₂O₂ with the lowest energy barrier, enhancing the thermodynamic favorability of this pathway. Furthermore, the subsequent generation of •OH proceed with a relatively low activation energy barrier over the Cu single-atom sites, indicating a kinetically favorable process. Therefore, based on the experimental results (Figure 2a) and charge density difference analysis (Figure S19-S20), the presence of N vacancies facilitated the oxidation of glucose, while the Cu single-atom sites promote the formation of •OH radicals with enhanced reaction kinetics. DFT calculations revealed that the embedding of Cu single-atom significantly reduces the reaction energy barrier in photocatalytic water splitting, thereby enhancing the reaction

kinetics. The rate-limiting step involved the cleavage of the O–H bond, which serves as the key intermediate transition state in the process (Figure 4e and S24).”

Figure 4. DFT of Cu/CN. The energy distribution corresponding to different transition states in (c) glucose conversion, (d) generation and conversion of H₂O₂, and (e) water splitting.

Figure S21. HPLC results of the glucose consumption.

Figure S22. Proposed reaction mechanism of glucose oxidation on CN₇₀₀ (a) and Cu/CN (b).

Figure S23. Proposed reaction mechanism of generation and conversion of H₂O₂ on CN₇₀₀ (a) and Cu/CN (b).

Figure S24. Proposed reaction mechanism of water splitting on CN₇₀₀ (a) and Cu/CN (b).

2. The manuscript claims that •OH and superoxide radicals •O₂⁻ are generated through photocatalytic pathways. However, the evidence supporting this claim is not sufficiently detailed. Given that glucose oxidation to glucuronic acid can lower the microenvironment's pH, which may enhance mimics enzyme activity of Cu/CN for ROS production, it is crucial to provide clear and direct evidence that •OH and •O₂⁻ are indeed produced via photocatalytic mechanisms rather than other pathways.

Response: Thanks for the reviewer's valuable comment. Following your suggestion, additional experiments were supplemented to fully elucidate that the ROS were produced by photocatalysis in this system.

The products of photocatalytic glucose consumption were analyzed by HPLC. The main product was arabinose, and no glucuronic acid was found (**Figure S21**). In order to prove clear that •OH and •O₂⁻ are indeed produced via photocatalytic mechanisms rather than other pathways, we carried out in situ EPR analysis, through lighting for a certain amount of time, and then avoiding light for a certain amount of time, cycle operation. The in situ EPR results and semi-quantitative statistical analysis are shown in **Figure S13-S14**, •OH and •O₂⁻ are increased under light, but not in the dark, it is proved that the free radical is produced by photocatalysis in this system. ROS are hyperactive and can not be scavenged after capture, ROS scavenging analysis can not be performed by in situ EPR.

The relevant description and discussion were supplemented in the revised manuscript main text (Page 11 and 14) and also shown below:

“To elucidate the photoswitch action, we performed in situ EPR analysis of Cu/CN under cyclic illumination, alternating between light exposure and dark intervals. The in situ EPR results and semi-quantitative statistical analysis (Figure S13-S14) demonstrated the light-dependent generation of $\bullet\text{OH}$ and $\bullet\text{O}_2^-$, with no ROS observed under dark conditions. Such results indicated that the generation of ROS in this system can be controlled by photoswitch.”

“The products of photocatalytic glucose consumption were analyzed by HPLC. The determined main product was arabinose, and there was no glucuronic acid found (Figure S21).”

Figure S13. (a) In situ EPR detection of $\bullet\text{OH}$ produced by Cu/CN in aqueous glucose solution under light-dark cycles. (b) Semi-quantitative statistical analysis was performed according to the peak height of EPR in (a).

Figure S14. (a) In situ EPR detection of $\bullet\text{O}_2^-$ produced by Cu/CN in methanol solution under light-dark cycles. (b) Semi-quantitative statistical analysis was performed according to the peak height of EPR in (a).

3. It is necessary to provide the ability of Cu/CN to clear ROS under different pH conditions. Can Cu/CN clear ROS in an acidic environment induced by glucuronic acid?

Response: Thanks for the reviewer’s valuable comment. Following your suggestion, additional experiments were supplemented to evaluate that the ability of Cu/CN to clear ROS under different pH conditions.

To further elucidate the ROS scavenging behavior of Cu/CN, we conducted additional experiments using DPPH and ABTS assays across varying pH conditions. As shown in **Figures S15-S16**, Cu/CN exhibits marginally enhanced ROS scavenging activity at lower pH values. The products of photocatalytic glucose consumption were analyzed by HPLC. The main product was arabinose, and no glucuronic acid was found (**Figure**

S21). Therefore, our Cu/CN is not a simple glucose oxidase-like enzyme. The photocatalytic glucose oxidation by Cu/CN mechanism differs fundamentally from enzymatic oxidation by glucose oxidase, as evidenced by HPLC and DFT.

The relevant description and discussion were supplemented in the revised manuscript main text (Page 11) and also shown below:

“To further explicate the ROS scavenging behavior of Cu/CN, we conducted DPPH and ABTS assays across varying pH conditions (3.0-7.0). As shown in Figure S15-S16, Cu/CN exhibited marginally enhanced ROS scavenging activity at lower pH. Furthermore, intracellular ROS scavenging investigation using RAW 264.7 murine macrophages confirmed that Cu/CN effectively eliminates excess ROS in dark conditions, indicating its light-independent antioxidant activity (Figure S17).”

Figure S15. ROS-scavenging efficiency of Cu/CN at different pH determined by DPPH method.

Figure S16. ROS-scavenging efficiency of Cu/CN at different pH determined by ABTS method.

4. The manuscript reports over 99.9% antibacterial activity and effective biofilm inhibition. However, the photocatalytic efficiency under different light intensities and wavelengths is not thoroughly investigated. Could the authors provide data on the photocatalytic performance under different light conditions (e.g., different wavelengths, intensities) to better understand the practical applicability of this system?

Response: Thanks for the reviewer’s valuable comment. Following your suggestion, additional experiments were supplemented to better understand the practical applicability of this system.

We have supplemented the activity experiments of Cu/CN photocatalysis for glucose

consumption under different wavelengths and light intensities. From the results, the activity of the Cu/CN is proportional to the UV-vis absorption, and the photocatalytic activity exhibits a positive correlation with light intensity. (**Figure S9-10**).

The relevant description and discussion were supplemented in the revised manuscript main text (Page 10) and also shown below:

“In addition, from the investigation of Cu/CN photocatalytic consumption of glucose under different wavelengths and light intensities, the photocatalytic activity of the Cu/CN exhibited positive correlations with both its UV-vis absorption capacity and the light intensity (Figure S9-10).”

Figure S9. Glucose consumption activity of Cu/CN under different wavelength (0.2 W/cm²).

Figure S10. Glucose consumption activity of Cu/CN under different light intensities.

5. The manuscript mentions that Cu/CN scavenges excess ROS in the dark, reducing inflammation. However, the exact mechanism by which Cu/CN scavenges ROS in the absence of light is not clearly explained. Could the authors provide more detailed experimental evidence or mechanistic insights into this process?

Response: Thanks for the reviewer’s valuable comment. Following your suggestion, additional experiments were supplemented to fully elucidate the detailed mechanistic pathways.

It can be seen from XPS (**Figure 3d**) and EXAFS (**Figure 3e**) that the Cu in Cu/CN exists in mixed valence states of +1 and +2. At the same time, XPS analysis of Cu 2p in Cu/CN after ROS scavenging revealed that the Cu²⁺ content increased from 29.1% to 47.6% (**Figure S18**). It shows that there is a Cu(+1)/Cu(+2) redox cycle in the process

of photocatalysis and scavenging ROS reaction. (*Nature Communications*, 2023, 14, 4583; *Nature Communications*, 2023, 14, 6741; *Nature Communications*, 2025, 16, 3202)

The relevant description and discussion were supplemented in the revised manuscript main text (Page 12) and also shown below:

“XPS analysis of Cu 2p in Cu/CN after ROS scavenging experiment revealed that the Cu²⁺ content increased from 29.1% to 47.6% (Figure S18), indicates there is a Cu(+1)/Cu(+2) redox cycle in the photocatalysis and scavenging processes.”

Figure S18. Cu 2p XPS spectra of Cu/CN before and after scavenge ROS during dark reaction.

Reviewer #2 (Remarks to the Author):

This manuscript "Cu Single-Atom Embedded g-C₃N₄ nanosheets Rehabilitate Multidrug-Resistant Bacteria Infected Diabetic Wounds via Photoswitchable Cascade Reaction" presents the preparation of a rational designed photocatalyst and their application in bacteria infected diabetic wounds therapy through programmed reactions. It is an interesting exploration to use multifunctional single atom catalysts for biomedical applications. This photocatalyst could consume the excess glucose and eliminate bacteria at diabetic wounds under light irradiation, and clear excess ROS without light to reduce inflammation and promote wound healing. In vitro studies demonstrate over 99.9% antibacterial efficacy and strong anti-biofilm ability towards multidrug resistant bacteria strains. In vivo animal studies exhibited good biological safety and excellent therapeutic efficacy. The major findings of this work are intriguing and meaningful, with relatively sufficient theoretical and experimental exploration. I recommend its acceptance for publication in this journal after minor revision. My comments are listed below:

1) The title mentions the use of "g-C₃N₄ nanosheets", but this term is not mentioned or explained in the main text. It is recommended that the author check their contents.

Response: Thanks for the reviewer's suggestion. We have rechecked the entire manuscript and made corrections. The revised manuscript has been updated accordingly.

(Please see the revised manuscript)

2) Why nitrogen vacancies of g-C₃N₄ could enhance the oxidation of glucose? The authors should provide more clear explanation and evidence.

Response: Thanks for the reviewer's questions. As can be seen from **Figure 2a**, the activity of the catalyst increases with increasing N vacancies. At the same time, it can be seen from DFT calculation that the adsorption energy of glucose at the nitrogen vacancy is lower, indicating that glucose is more likely to be adsorbed at the nitrogen vacancy, which is more conducive to the reaction. Also shown in **Figures S19-20** are the electron accumulation (yellow) and electron depletion (blue) of glucose at non-nitrogen-vacancy sites, indicating that the presence of nitrogen vacancies oxidizes glucose more efficiently.

Figure 2. Photoelectrochemical behaviours of modified CN. (a) Photocatalytic properties of different catalysts for glucose consumption.

Figure S20. Top and side views of the Charge Density Difference Analysis for the glucose-N. The yellow and the cyan areas represent charge accumulation and depletion, respectively.

Figure S21. Top and side views of the Charge Density Difference Analysis for the glucose-NV. The yellow and the cyan areas represent charge accumulation and depletion, respectively.

3) Some important details are missed in their biological study. What is the concentration of Cu/CN used in the cell experiments? What is the concentration of Cu/CN used for in vivo treatment? How many times and how long was Cu/CN used for in vivo treatment experiments?

Response: Thanks for the reviewer's questions. We have rechecked the entire manuscript and made corrections. To comprehensively evaluate the material's performance, we have supplemented experimental details in four key areas: (1) ROS-scavenging ability evaluation, (2) Products analysis of photocatalytic glucose consumption, (3) In vitro antibiofilm tests, and (4) In vitro cell migration assay of Cu/CN. Furthermore, we have enhanced the methodological details for: (1) Hypoglycemic ability of CN and Cu/CN, (2) In vitro antibacterial performance of CN and Cu/CN, (3) In vitro cytotoxicity assay, (4) Diabetic chronic wound healing experiment, and (5) In vitro ROS scavenging activities of Cu/CN.

The relevant description was supplemented in the revised manuscript supporting information (Pages 3-8).

4) It is necessary to supplement the cytotoxicity study of Cu/CN at different concentrations.

Response: Thanks for the reviewer's suggestion. We supplemented cytotoxicity experiments with different concentrations of Cu/CN and the results as **Figure S25** showed that Cu/CN remained cell friendly and non-toxic at concentrations up to 1000 µg/mL. (Please see Page 19 in the revised manuscript and also shown below)

“Notably, Cu/CN showed no significant cytotoxicity even at elevated concentrations up to 1000 µg/mL (Figure S25).”

Figure S25. Cell viability of 3T3 cells incubated with Cu/CN at different concentrations.

5) P18, line9, “requires NIR irradiation for effective activity”, please verify this content.

Response: Thanks for the reviewer's suggestion. We have rechecked the entire manuscript and made corrections. The revised manuscript has been updated accordingly. (Please see Page 19 in the revised manuscript and also shown below)

“In contrast, wound healing was significantly delayed in dark conditions, with only 37.3±0.8% closure achieved for Cu/CN treated group at day 14 (Figure S27), demonstrating its therapeutic efficacy is light-dependent.”

6) The text contains multiple instances where there is no space before parentheses and inconsistent font usage. It is recommended that the author thoroughly review and revise the entire manuscript to address these article format issues.

Response: Thanks for the reviewer's suggestion. We have rechecked the entire manuscript and made corrections. The manuscript underwent professional language polishing by native English-speaking scholars. The revised manuscript has been updated accordingly.

(Please see the revised manuscript)

Reviewer #3 (Remarks to the Author):

The work present by Sun et al. is novel to some extent, robust and presents significant fundamental and applicative implications and advancements. Most of the conclusions and claims are well supported by data, experiments and analysis. It is also relevant and important to the field of study. The quality of the work satisfies the high standards desired for consideration in Nat. Commun. Thus, I would recommend reconsidering this work for acceptance after the following minor changes and suggestions are addressed.

There are several grammatical and sentence construction errors at various places in the manuscript. Please carefully proofread and rectify those.

Response:

Thanks for the reviewer's valuable comment. Following your suggestion, we have rechecked the entire manuscript and made corrections. The manuscript underwent professional language polishing by native English-speaking scholars. The revised manuscript has been updated accordingly.

(Please see the revised manuscript)

The prominent peak in FTIR at 2175 cm⁻¹ is not that prominent.. also it appears to be present in a similar fashion in all the annealed samples. I fail to identify a significant change for the different thermally annealed vs. control. Please explain/justify

Response: Thanks for the reviewer's valuable comment. Following your suggestion, after baseline correction of the FTIR raw data, we observed that the intensity of the absorption peak at 2175 cm⁻¹ increased with rising annealing temperature across different materials (**Figure 1f**).

(Please see Page 7 in the revised manuscript and also shown below)

Figure 1. Composition characterization of CN. FT-IR spectra (f) of CN with different nitrogen vacancy concentrations.

Please provide elemental ratio (C/N) and Raman peak shifts (particularly corresponding to the triazine rings) to further prove/probe nitrogen vacancies in different thermally etched samples.

Response: Thanks for the reviewer’s valuable comment. Following your suggestion, additional experiments were supplemented to prove the nitrogen vacancies in different samples.

To further prove the nitrogen vacancies in different samples, we performed elemental analysis and Raman spectroscopy. The increased C/N ratio from 0.65 to 0.69 supports the formation of nitrogen vacancies (**Figure S4**), and the decreased Raman peaks indicates the structural unit of the CN is destroyed with the increasing annealing temperature (**Figure S5**).

The relevant description and discussion were supplemented in the revised manuscript main text (Page 6) and also shown below:

“To further prove the N vacancies in different samples, we performed elemental analysis and Raman spectroscopy. The increased C/N ratio from 0.65 to 0.69 supported the formation of N vacancies (Figure S4), and the decreased Raman peak intensities indicates the structural unit of the CN is compromised with the increase of annealing temperature (Figure S5).”

Figure S4. The C/N ratio characterized by elemental analysis of CN₅₅₀, CN₆₀₀, CN₆₅₀, CN₇₀₀ and CN₇₅₀.

Figure S5. The Raman spectra of CN₅₅₀, CN₆₀₀, CN₆₅₀, CN₇₀₀ and CN₇₅₀ excited with a NIR laser, 785 nm.

Does the N-vacancies lead to a red shift in UV-Vis abs? -- Please check and comment.

Response: Thanks for the reviewer's questions. The N-vacancy causes a red shift in the UV-vis absorption, which can be seen in **Figure 1d** as a slight red shift with increasing N-vacancy. The forbidden band width of the material is determined by the Tauc curve, as **Figure 2f**, the forbidden band width narrows with the increase of N-vacancy. The band gap is also calculated by PBE (**Figure S6**). The band gap becomes narrow in the presence of N vacancy.

Figure 1. Composition characterization of CN. UV-vis spectra of CN with different nitrogen vacancy concentrations.

Figure 2. Photoelectrochemical behaviours of modified CN. (f) Schematic illustration of band structure for the different catalysts.

Figure S6. PBE of (a) CN, (b) CN₇₀₀, and (c) Cu/CN.

Please conduct TRPL measurements to analyze the charge-carrier lifetime for different samples.

Response: Thanks for the reviewer's valuable comment. Following your suggestion, additional experiments were supplemented to analyze the charge-carrier lifetime for different samples.

To analyze the charge-carrier lifetime for different samples, we performed TRPL measurements. The results are presented in **Figures S7** and **Table S5**. The results show that the fluorescence lifetime of CN increases from 6.54 to 12.45 ns with the increase of annealing temperature, and the fluorescence lifetime of CN increases with the increase of N vacancy, the N vacancy increases the utilization ability of photogenerated carriers of the material, and the loading of Cu single-atom also increases the fluorescence lifetime of the material to 15.06 ns. The revised manuscript has been updated accordingly.

The relevant description and discussion were supplemented in the revised manuscript main text (Page 9) and also shown below:

“To analyze the charge-carrier lifetime for different samples, we performed time-resolved photoluminescence (TRPL) spectroscopy. As presented in Figure S7 and Table S5, the fluorescence lifetime of CN increases from 6.54 to 12.45 ns as the annealing temperature rises. This trend correlates with the increase of N vacancies, which promotes the photogenerated carriers utilization capability of CN.”

Figure S7. (a) The TRPL spectra of CN₅₅₀, CN₆₀₀, CN₆₅₀, CN₇₀₀ and CN₇₅₀. (b) The TRPL spectra of CN₇₀₀ and Cu/CN.

Table S5. Average lifetimes of photogenerated charge carriers in CN₅₅₀, CN₆₀₀, CN₆₅₀, CN₇₀₀, CN₇₅₀ and Cu/CN.

Model	ExpDec3					
Equation	$y = A_1 \cdot \exp(-x/\tau_1) + A_2 \cdot \exp(-x/\tau_2) + A_3 \cdot \exp(-x/\tau_3) + y_0$					
Catalyst	CN ₅₅₀	CN ₆₀₀	CN ₆₅₀	CN ₇₀₀	CN ₇₅₀	Cu/CN
y_0	0.00041	-0.0008	-0.00194	-0.00389	-0.00214	-0.00568
A_1	0.63514	0.69787	0.75869	0.76205	0.74971	0.73335
τ_1	0.92878	1.03703	0.8402	0.95102	0.96847	0.97981
A_2	0.32827	0.29858	0.25565	0.2564	0.27101	0.27147
τ_2	3.54217	4.04169	3.55719	3.99582	4.00381	4.128
A_3	0.04603	0.03621	0.02814	0.02748	0.03248	0.03151
τ_3	15.85249	21.58852	24.99563	31.86752	25.74288	35.21762
τ_{ave}	6.53972 ns	8.29735 ns	9.48876 ns	12.45349 ns	10.03749 ns	15.06038 ns

The authors didn't mention about the quantitative Cu loading in the Cu/CN sample in the manuscript (only in SI). Please include it in the discussion combining results from ICP/XPS and EDS.

Response: Thanks for the reviewer's valuable comment. Following your suggestion, additional experiments were supplemented to evaluate that the uniform distribution of C, N and Cu elements in Cu/CN nanosheets.

We have supplemented the EDS mapping test results. The ICP results are also explained in the main text. The energy-dispersive X-ray spectra (EDS) displayed in **Figure 3k** clearly shows the uniform distribution of carbon (C), nitrogen (N) and copper (Cu) elements in Cu/CN nanosheets. The actual content of Cu was 1 wt% as determined by inductively coupled plasma optical emission spectroscopy (ICP-OES).

The relevant description and discussion were supplemented in the revised manuscript main text (Page 11) and also shown below:

“And the energy-dispersive X-ray spectroscopy (EDS) elemental mapping confirmed the homogeneous distribution of constituent elements (C, N) and embedded copper (Cu) across the Cu/CN nanosheets (Figure 3k). The actual content of Cu was 0.99 ± 0.01 wt% as determined by inductively coupled plasma optical emission spectroscopy (ICP-OES).”

Figure 3. Structural characterizations of Cu/CN. (k) EDS mapping of Cu/CN.

The authors should provide HAADF-STEM images to show the single atom dispersion of Cu in the CN matrix.

Response: Thanks for the reviewer’s valuable comment. Following your suggestion, additional experiments were supplemented to prove that the single atom dispersion of Cu in the CN matrix.

We supplemented the AC-HAADF-STEM and observed that the Cu on the Cu/CN surface was dispersed into single-atom (**Figure 3j**).

The relevant description and discussion were supplemented in the revised manuscript main text (Page 11) and also shown below:

“Furthermore, the observation by aberration-corrected high-angle annular dark-field scanning transmission electron microscopy (AC-HAADF-STEM) also verified the atomically dispersed Cu on Cu/CN nanosheets (Figure 3j).”

Figure 3. Structural characterizations of Cu/CN. (j) AC-HAADF-STEM image of Cu/CN. The atomically dispersed Cu was highlighted by yellow circles.

The experimental/methodology section of the paper is not elaborate enough—many of the characterization and details are missing. The authors should provide all the information in detail to ensure reproducibility. Many characterization details and specifications are missing (please check carefully and provide the necessary info). How were the materials used in the bandages in the animal model would healing experiments?

Response: Thanks for the reviewer’s questions. We have rechecked the entire manuscript and made corrections. To comprehensively evaluate the material’s performance, we have supplemented experimental details in four key areas: (1) ROS-scavenging ability evaluation, (2) Products analysis of photocatalytic glucose consumption, (3) In vitro antibiofilm tests, and (4) In vitro cell migration assay of Cu/CN. Furthermore, we have enhanced the methodological details for: (1) Hypoglycemic ability of CN and Cu/CN, (2) In vitro antibacterial performance of CN and Cu/CN, (3) In vitro cytotoxicity assay, (4) Diabetic chronic wound healing experiment, and (5) In vitro ROS scavenging activities of Cu/CN.

The relevant description was supplemented in the revised manuscript supporting information (Pages 3-8).

Reviewer #4 (Remarks to the Author):

The study presents a promising strategy for diabetic wound management with robust in vitro and in vivo results. Addressing the above points will enhance the manuscript more readable.

1、The title emphasizes "photoswitchable cascade reaction," but the mechanism of light-switching (on/off control) needs clearer experimental validation. The abstract is comprehensive but could benefit from conciseness, focusing on the most impactful findings.

Response: Thanks for the reviewer's valuable comment. Following your suggestion, additional experiments were supplemented to prove that the mechanism of light-switching (on/off control).

In order to prove the action mechanism of the photoswitch, we carried out in situ EPR analysis, through lighting for a certain amount of time, and then avoiding light for a certain amount of time, cycle operation. The in situ EPR results and semi-quantitative statistical analysis are shown in **Figure S13-14**, $\bullet\text{OH}$ and $\bullet\text{O}_2^-$ are increased under light, but not in the dark, it is proved that the generation of free radicals can be controlled by photoswitch in this system. ROS are hyperactive and can not be scavenged after capture, ROS scavenging analysis can not be performed by in situ EPR. However, we performed free radical scavenging experiments with DPPH, ABTS, and cell experiments, such as **Figure S15-16**, which were able to demonstrate the free radical scavenging properties of the material in the dark. The revised manuscript has been updated accordingly.

The relevant description and discussion were supplemented in the revised manuscript main text (Page 11) and also shown below:

“To elucidate the photoswitch action, we performed in situ EPR analysis of Cu/CN under cyclic illumination, alternating between light exposure and dark intervals. The in situ EPR results and semi-quantitative statistical analysis (Figure S13-S14) demonstrated the light-dependent generation of $\bullet\text{OH}$ and $\bullet\text{O}_2^-$, with no ROS observed under dark conditions. Such results indicated that the generation of ROS in this system can be controlled by photoswitch.”

The relevant changes have been made to the abstract section in the revised manuscript main text (Page 2) and also shown below:

“Under visible light illumination, the N vacancy exist in g-C₃N₄ (CN) significantly enhances photocatalytic glucose oxidation to regulate the hyperglycemia condition at diabetic wound sites, and the atomically dispersed Cu promotes the generation of $\bullet\text{OH}$ and $\bullet\text{O}_2^-$ to efficiently eliminate MDR bacteria (>99.9%). Under dark conditions, excess ROS are scavenged by Cu/CN, reducing inflammation of wounds and promoting polarization of M2 macrophages. Serum biochemical and vital organs histopathological analyses after 14 days of treatment confirmed the exceptional biosafety profile of Cu/CN.”

Figure S13. (a) In situ EPR detection of $\bullet\text{OH}$ produced by Cu/CN in aqueous glucose solution under light-dark cycles. (b) Semi-quantitative statistical analysis was performed according to the peak height of EPR in (a).

Figure S14. (a) In situ EPR detection of $\bullet\text{O}_2^-$ produced by Cu/CN in methanol solution under light-dark cycles. (b) Semi-quantitative statistical analysis was performed according to the peak height of EPR in (a).

Figure S15. ROS-scavenging efficiency of Cu/CN at different pH determined by DPPH method.

Figure S16. ROS-scavenging efficiency of Cu/CN at different pH determined by ABTS method.

Figure S17. Intracellular ROS-scavenging performance of Cu/CN.

2、 The synergistic effect of nitrogen vacancies and Cu single-atom embedding is a key innovation. However, the introduction should more explicitly contrast this approach with prior studies using defect engineering or single atoms alone, emphasizing the novelty of combining both strategies.

Response: Thanks for the reviewer’s valuable comment. Following your suggestion, additional descriptions were supplemented to emphasize the novelty of combining both strategies.

The relevant description was supplemented in the revised manuscript main text (Page 4) and also shown below:

“Recent studies have employed several single-atom catalysts for photocatalytic antibacterial applications, including Ag⁴⁷, Zn⁴⁸, and Cu⁴⁹ single-atoms. However, none of these works explored the synergistic combination with N vacancies to enhance their photocatalytic activity.”

3、 The synthesis of Cu/CN (e.g., calcination temperatures, Cu loading ratios) requires more precise parameters to ensure reproducibility. Additionally, HAADF-STEM or AC-STEM images are necessary to confirm the atomic dispersion of Cu, as XRD alone cannot rule out nanoparticle formation.

Response: Thanks for the reviewer’s valuable comment. Following your suggestion, additional experiments were supplemented to confirm that the atomic dispersion of Cu in the CN.

We supplemented the AC-HAADF-STEM and observed that the Cu on the Cu/CN surface was dispersed into single-atom (**Figure 3j**).

The relevant description and discussion were supplemented in the revised manuscript main text (Page 11) and also shown below:

“Furthermore, the observation by aberration-corrected high-angle annular dark-field scanning transmission electron microscopy (AC-HAADF-STEM) also verified the atomically dispersed Cu on Cu/CN nanosheets (Figure 3j).”

Figure 3. Structural characterizations of Cu/CN. (j) AC-HAADF-STEM image of Cu/CN. The atomically dispersed Cu was highlighted by yellow circles.

4、 The claim that Cu/CN scavenges excess ROS in the dark lacks direct evidence. Quantifying ROS levels (e.g., via fluorescence probes) under dark conditions would strengthen this conclusion.

Response: Thanks for the reviewer's suggestion. It can be seen from XPS (**Figure 3d**) and EXAFS (**Figure 3e**) that the Cu in Cu/CN exists in mixed valence states of +1 and +2. At the same time, XPS analysis of Cu 2p in Cu/CN after ROS scavenging experiment revealed that the Cu²⁺ content increased from 29.1% to 47.6% (**Figure S18**), indicating there is a Cu(+1)/Cu(+2) redox cycle in the photocatalysis and scavenging processes. (*Nature Communications*, 2023, 14, 4583; *Nature Communications*, 2023, 14, 6741; *Nature Communications*, 2025, 16, 3202)

ROS are hyperactive and can not be scavenged after capture, ROS scavenging analysis can not be performed by in situ EPR. However, we performed free radical scavenging experiments with DPPH, ABTS, and cell experiments, such as **Figure S15-17**, which were able to demonstrate the free radical scavenging properties of the material in the dark. The revised manuscript has been updated accordingly.

The relevant description and discussion were supplemented in the revised manuscript main text (Pages 11-12) and also shown below:

“To further explicate the ROS scavenging behavior of Cu/CN, we conducted DPPH and ABTS assays across varying pH conditions (3.0-7.0). As shown in Figure S15-S16, Cu/CN exhibited marginally enhanced ROS scavenging activity at lower pH. Furthermore, intracellular ROS scavenging investigation using RAW 264.7 murine macrophages confirmed that Cu/CN effectively eliminates excess ROS in dark conditions, indicating its light-independent antioxidant activity (Figure S17).”

“XPS analysis of Cu 2p in Cu/CN after ROS scavenging experiment revealed that the Cu²⁺ content increased from 29.1% to 47.6% (Figure S18), indicating there is a Cu(+1)/Cu(+2) redox cycle in the photocatalysis and scavenging processes.^{57, 58, 59”}

5、 While bactericidal rates are impressive, statistical significance markers (e.g., p-values) are missing in Figure 5b/d. Clarify the sample size and replicate numbers for bacterial viability assays.

Response: Thanks for the reviewer’s valuable comment. Following your suggestion, additional descriptions were supplemented to incorporate statistical significance markers.

The relevant description was supplemented in the revised manuscript main text (Page 18) and also shown below:

*“(P values are calculated by the one-way ANOVA method, *** $p < 0.001$; **** $p < 0.0001$). The results were expressed as the mean of at least five replicates \pm SD (standard deviation).”*

6、 Hemolysis and cell viability assays are well-presented, but long-term cytotoxicity (e.g., 7-day exposure) and in vivo toxicity (e.g., histopathology of major organs) should be included to address potential safety concerns.

Response: Thanks for the reviewer’s valuable comment. Following your suggestion, additional experiments were supplemented to address potential safety concerns.

To address potential safety concerns, we supplemented comprehensive toxicity assessments by analyzing serum biochemical markers-including alanine aminotransferase (ALT), aspartate aminotransferase (AST), creatinine (CRE), and blood urea nitrogen (BUN)-as well as histopathological examinations of major organs (heart, liver, spleen, lung, and kidney) following 14 days treatment in mice. Biochemical results demonstrated that two weeks Cu/CN treatment elicited no statistically significant alterations in mice hematological parameters and hepatic/renal function indices (**Figure S29**). Histopathological analysis also revealed there was no evidence of structural damage or pathological abnormalities in vital organs (**Figure S30**). These collective findings confirmed that Cu/CN possesses an exceptional in vivo biosafety profile for therapeutic applications.

The relevant description and discussion were supplemented in the revised manuscript main text (Pages 22-23) and also shown below:

“To evaluate the biosafety of Cu/CN, we conducted comprehensive toxicity assessments by analyzing serum biochemical markers-including alanine aminotransferase (ALT), aspartate aminotransferase (AST), creatinine (CRE), and blood urea nitrogen (BUN)-as well as histopathological examinations of major organs (heart, liver, spleen, lung, and kidney) following 14 days treatment in mice. Biochemical results demonstrated that two weeks Cu/CN treatment elicited no statistically significant alterations in mice hematological parameters and hepatic/renal function indices (Figure S29). Histopathological analysis also revealed there was no evidence of structural damage or pathological abnormalities in vital organs (Figure S30). These collective findings confirmed that Cu/CN possesses an exceptional in vivo biosafety profile for therapeutic applications.”

Figure S29. Serum biochemical markers (ALT, AST, BUN, CRE) were quantified post-treatment (day 14) across control and Cu/CN groups.

Figure S27. H&E staining of major organs of mice after different treatments.

7、 The adsorption energy calculations for glucose and H_2O_2 are insightful. However, provide additional computational evidence (e.g., reaction pathways, activation barriers) to support the proposed cascade mechanism.

Response: Thanks for the reviewer's valuable comment. Following your suggestion, additional experiments and DFT calculations were supplemented to fully elucidate the detailed cascade mechanistic pathways.

The products of photocatalytic glucose consumption were analyzed by HPLC. The determined main product was arabinose, and there was no glucuronic acid found (**Figure S21**). So the complete equation would be:

The oxidation reaction occurring at the valence band position is

The reduction reaction of H_2O_2 generated by single electron transfer of O_2 at the conduction band position is

DFT calculations were performed for both reactions (2) and (3), with the corresponding energy profiles presented in newly supplemented **Figure 4c-d** and **Figure S22-23**. The upstream step is endothermic and non-spontaneous, whereas the downstream step is

exothermic and spontaneous. The step with the highest endothermic energy corresponds to the rate-determining step (RDS) of the overall reaction. A lower energy barrier for the RDS leads to enhanced catalytic activity. The presence of Cu single-atom catalysts significantly reduces the energy barrier of the rate-determining step in reaction (2), thereby facilitating the reaction kinetics. For reaction (3), the Cu single-atom catalyst enables the formation of H₂O₂ with the lowest energy barrier, enhancing the thermodynamic favorability of this pathway. Furthermore, the subsequent generation of •OH radicals proceed with a relatively low activation energy barrier over the Cu single-atom sites, indicating a kinetically favorable process. Therefore, based on the experimental results (**Figure 2a**) and charge density difference analysis (**Figure S19-S20**), the presence of N vacancies facilitates the oxidation of glucose, while the Cu single-atom sites promote the formation of •OH radicals with enhanced reaction kinetics.

The DFT calculation results reveal that the introduction of Cu single-atom significantly reduces the reaction energy barrier in photocatalytic water splitting, thereby enhancing the reaction kinetics. The rate-limiting step involves the cleavage of the O–H bond, which serves as the key intermediate transition state in this process (**Figure 4e** and **S24**). The relevant description and discussion were supplemented in the revised manuscript main text (Page 14) and also shown below:

“The products of photocatalytic glucose consumption were analyzed by HPLC. The determined main product was arabinose, and there was no glucuronic acid found (Figure S21). The complete equation would be:

The oxidation reaction occurring at the valence band position was

The reduction reaction of H₂O₂ generated by single electron transfer of O₂ at the conduction band position was

DFT calculations were performed for both reactions, with the corresponding energy profiles presented in Figure 4c-d and Figure S22-S23. The upstream step was endothermic and non-spontaneous, whereas the downstream step is exothermic and spontaneous. The step with the highest endothermic energy corresponded to the rate-determining step (RDS) of the overall reaction. A lower energy barrier for the RDS leads to enhanced catalytic activity. The presence of Cu single-atom catalysts significantly reduced the energy barrier of the rate-determining step in reaction (2), thereby facilitating the reaction kinetics. For reaction (3), the Cu single-atom catalyst enabled the formation of H₂O₂ with the lowest energy barrier, enhancing the thermodynamic favorability of this pathway. Furthermore, the subsequent generation of •OH proceed with a relatively low activation energy barrier over the Cu single-atom sites, indicating a kinetically favorable process. Therefore, based on the experimental results (Figure 2a) and charge density difference analysis (Figure S19-S20), the presence of N vacancies facilitated the oxidation of glucose, while the Cu single-atom sites promote the formation of •OH radicals with enhanced reaction kinetics. DFT calculations

revealed that the embedding of Cu single-atom significantly reduces the reaction energy barrier in photocatalytic water splitting, thereby enhancing the reaction kinetics. The rate-limiting step involved the cleavage of the O–H bond, which serves as the key intermediate transition state in the process (Figure 4e and S24).”

Figure 4. DFT of Cu/CN. The energy distribution corresponding to different transition states in (c) glucose conversion, (d) generation and conversion of H₂O₂, and (e) water splitting.

Figure S21. HPLC results of the glucose consumption.

Figure S22. Proposed reaction mechanism of glucose oxidation on CN₇₀₀ (a) and Cu/CN (b).

Figure S23. Proposed reaction mechanism of generation and conversion of H₂O₂ on CN₇₀₀ (a) and Cu/CN (b).

Figure S24. Proposed reaction mechanism of water splitting on CN₇₀₀ (a) and Cu/CN (b).

8、The discussion should address scalability (e.g., cost of Cu/CN synthesis) and practical challenges (e.g., light penetration depth in human tissue) for translational applications.

Response: Thanks for the reviewer’s valuable comment. Following your suggestion, additional descriptions were supplemented to address scalability and practical challenges for translational applications.

The relevant description was supplemented in the revised manuscript main text (Page 25) and also shown below:

“The Cu/CN photocatalyst presents a multifaceted therapeutic platform with distinct advantages for biomedical applications. The Cu/CN is synthesized through facile thermal polycondensation and impregnation processes, using cost-effective commercially available precursors. The material’s properties can be precisely tuned through defect engineering, enabling customized electronic environment for optimized performance. The visible light-responsive nature of this photocatalyst renders it particularly suitable for superficial tissue applications, as the penetration depth of these wavelengths optimally matches dermal thicknesses, making it ideal for managing cutaneous pathologies such as diabetic wounds. This system demonstrates multiple therapeutic efficacies by simultaneously addressing localized hyperglycemia, providing

robust antibacterial activity, and regulating inflammation, which synergistically enhance diabetic wound repair processes. Furthermore, as a functional nanomaterial, Cu/CN exhibits significant potential for integration into advanced delivery systems such as microneedle patches or nanovectors, to expand its clinical applicability.”

9、 Ensure all cited references (e.g., Ref. 6) are peer-reviewed and publicly available.

Response: Thanks for the reviewer’s valuable comment. All cited references in this work are peer-reviewed publications and publicly accessible. According to information from the journal editorial office, the formal issue, volume number, and page numbers for this article (Ref. 6, <https://doi.org/10.1016/j.fmre.2023.08.012>) are expected to be assigned by the end of September 2025.

Response to the reviewers' comments

Reviewer #1 (Remarks to the Author):

1. Although the authors have proved the production of arabinose rather than the glucuronic acid, the byproduct HCOOH is highly acidic that can also significantly reduce the pH value in the micro environment. In general, the POD activity of nanozyme can be activated at low pH value, therefore, it is hard to say the ROS production was triggered by light or the low pH value. The reviewer suggest the author to study the production of ROS (by ESR) under low pH value in the absence of light to clarify this point.

Response: Thank you for the valuable comments. As suggested, additional experiments were conducted to clarify the ROS production is induced by light rather than low pH.

The pH of both glucose and the H₂O₂ solution was adjusted to 5 and 3 individually using HCOOH. After adding Cu/CN and DMPO, the reaction mixture was incubated in the dark for 10 minutes. The EPR spectroscopy detected no ROS generation under those low-pH conditions without light irradiation, confirming the light-dependence ROS production in this system.

The relevant description and discussion were supplemented in the revised manuscript main text (Page 11) and also shown below:

“furthermore, detected no ROS generation under those low-pH conditions without light illumination (Figure S15).”

Figure S15. EPR spectroscopy of ROS produced in the dark by Cu/CN in glucose solution at different pH (pH=5 and 3) adjusted by HCOOH.

2. In the response section, The author claim that “The reduction reaction of H₂O₂ generated by single electron transfer of O₂ at the conduction band position was equation 3.” Equation 3 doesn’t seem right. In this reaction the superoxide should be firstly produced with one electron injection from the CB.

Response: We appreciate the reviewer’s constructive suggestion. Following your suggestion, we have supplemented the suggested reaction step to the revised version (Page 14) and also shown below:

3. Can the author provide more details about the TS in Figure 4c.

Response: Thanks for the reviewer’s valuable comment. Following your suggestion, more details about the TS in Figure 4c and Figure S22 were supplemented to fully elucidate the detailed mechanistic pathways.

The transition state (TS) corresponds to a critical configuration where the glucose molecule undergoes C-C bond elongation prior to methanol elimination, representing a metastable “bond-breaking yet unbroken” state. DFT calculations reveal that the C-C bond length increases from 1.54 Å (ground state) to 1.60 Å in the presence of Cu/CN, comparatively, CN₇₀₀ induces greater bond elongation to 1.62 Å under identical conditions (Figure S23). These results demonstrate that Cu/CN exhibits superior catalytic activity for hydroxymethyl elimination from glucose, as evidenced by the lower reaction energy barrier required to achieve the transition state compared to CN₇₀₀.

The above description and discussion were supplemented in the revised manuscript main text (Page 14), Figure 4c and Figure S22 were updated as shown below.

Figure 4. DFT of Cu/CN. The energy distribution corresponding to different transition states in (c) glucose conversio.

Figure S23. Proposed reaction mechanism of glucose oxidation on CN₇₀₀ (a) and Cu/CN (b).

Reviewer #2 (Remarks to the Author):

The present manuscript had been revised in an appropriate and disciplined manner. The revision increased the quality of the manuscript and hence the manuscript may be suitable for its publication in its present form.

Response: We sincerely appreciate the expert guidance and invaluable support.

Reviewer #3 (Remarks to the Author):

The authors have satisfactorily addressed all the concerns and queries raised by the reviewers through additional experiments and appropriate revisions. The comprehensive effort in responding to the all the referees' feedback has significantly enhanced the quality and clarity of the manuscript and its claims. I, therefore, recommend the acceptance and publication of the manuscript in its current form.

Response: We sincerely appreciate the expert guidance and invaluable support.

Reviewer #4 (Remarks to the Author):

After authors' carefully revision and answer, all questions are resolved and the manuscript is ready for publication.

Response: We sincerely appreciate the expert guidance and invaluable support.